



# Measurement of size dependent single scattering albedo of fresh biomass burning aerosols using the extinction-minus-scattering technique with a combination of cavity ring-down spectroscopy and nephelometry.

**Sujeeta Singh,[1] Marc N. Fiddler,[2] and Solomon Bililign[2,3]**

[1]Energy and Environmental Systems Department, North Carolina A&T State University, Greensboro, North Carolina, USA
[2]Department of Physics, North Carolina A&T State University, Greensboro, North Carolina, USA
[3]NOAA-ISET Center, North Carolina A&T State University, Greensboro, North Carolina, USA

*Correspondence to*: bililign@ncat.edu

**Abstract.** Biomass burning (BB) aerosols have a significant effect on regional climate, and represent a significant uncertainty in our understanding of climate change. Using a combination of cavity ring-down spectroscopy and integrating nephelometry, the single scattering albedo (SSA) and Ångstrom absorption exponent (AAE) were measured for several North American fuels. This was done for several particle diameters for the smoldering and flaming stage of white pine, red oak, and cedar combustion. Measurements were done over a wider wavelength range than any previous direct measurement of BB particles. While the offline sampling system used in this work shows promise, some changes in particle size distribution were observed, and a thorough evaluation of this method is required. The uncertainty of SSA was 6 %, with the truncation angle correction of the nephelometer being the largest contributor to error. While scattering and extinction did show wavelength dependence, SSA did not. SSA values ranged from 0.46 to 0.74, and were not uniformly greater for the smoldering stage than the flaming stage. SSA values changed with particle size, and not systematically so, suggesting the proportion of tar balls to fractal black carbon change with fuel type/state and particle size. SSA differences of 0.15−0.4 or greater can be attributed to fuel type or fuel state for fresh soot. AAE values were quite high (1.59−5.57), despite SSA being lower than is typically observed in wildfires. The SSA and AAE values in this work do not fit well with current schemes that relate these factors to the modified combustion efficiency of a burn. Combustion stage, particle size, fuel type, and fuel condition were found to have most significant effect on the intrinsic optical properties of fresh soot, though additional factors influence aged soot.

## 1. Introduction

Biomass burning (BB) is recognized as one of the largest sources of absorbing aerosols in the atmosphere (Bond et al.2013;Jacobson, 2014;Ramanathan and Carmichael, 2008;Moosmüller et al., 2009). Smoke from BB is composed of gaseous and aerosol constituents, including black carbon (BC), brown carbon (BrC), organic carbon (OC), and mineral dust; all of which have critical climate and health impacts. Global climate impacts of BB result from its truly massive contributions to aerosol optical depth over large areas and from secondary processes, such as cloud and ice nucleation, that can increase the radiative impact of the emissions. BB aerosols have significant impacts, not





only on local, but also on regional climate, air quality, and hydrological cycles (Alonso-Blanco et al.,
2014;Haywood et al., 2003;Haywood et al., 2008;Fu et al., 2012;Lin et al., 2013;Reid et al., 2013;Yen et al., 2013).

With an estimated total climate forcing of +1.1 W•m$^{-2}$, BC is the second most important human emission in terms of
its climate forcing in the present-day atmosphere; second to $CO_2$ (Bond et al., 2013). The impacts of wildfires are
mostly associated with short-term climate forcers, such as ozone and aerosols. Depending on surface albedo and the
relative amounts of OC and BC/BrC, BB smoke can heat or cool the atmosphere, provide condensation nuclei for
ice and water reduce visibility, and affect air quality. The recent estimate (IPCC, 2013) of biomass aerosol radiative
forcing is 50 % larger than earlier estimates.

In the atmosphere, aerosols dynamically change in complex ways. BC is initially produced during the combustion of
carbon-based fuels when oxygen is insufficient for complete combustion during BB (Bond et al., 2013;Bond,
2006;Bond and Bergstrom, 2006). The chemical composition and physical properties of particles then evolve during
their atmospheric lifetime due to condensation, oxidation reactions, etc. Soot is formed from organic precursors in
high temperatures and insufficient oxygen environments where volatiles and primary tars react to form secondary
tars to form polyaromatic hydrocarbons (PAH), which subsequently form soot particles by further agglomeration
and release of hydrogen (Nussbaumer, 2010).

A theoretical BC aging model was developed to account for three major stages of aging: aggregates of graphitic
spheres and primary tars freshly emitted from BB, aggregates becoming coated with condensable material, and BC
particles undergoing further hygroscopic growth (He et al., 2015). BB aerosols are subject to extensive chemical
processing in the atmosphere as they are exposed to sunlight, other pollutants like biogenic VOCs, and oxidants such
as ozone ($O_3$), hydroxyl radical (OH), and $NO_x$ ($NO+NO_2$). The timescale for these processes are quite short; on the
order of a few minutes to hours (Hennigan et al., 2011;Rudich et al., 2007;Saleh et al., 2013;Hemminger, 1999;Haan
et al., 1999;Cubison et al., 2011;Vakkari et al., 2014). Additionally, there is evidence of both loss and gain of
particle mass, and rapid atmospheric oxidation (Vakkari et al., 2014). While semi-volatile compounds condense
when they are cooled, as smoke is diluted, these compounds can revolatilize, which reduces aerosol mass (Robinson
et al., 2007).

As these physical and chemical changes take place, the optical properties of these particles are also altered.
Variations in optical properties of soot particles due to internal mixing in the atmosphere and aging remain highly
uncertain, hindering efforts to assess their impact on climate. Understanding the effect of aging on composition and
the commensurate optical property changes remains a challenge. Theoretical calculations are consistent with
measurements in extinction and absorption cross sections for fresh BC aggregates, but overestimate the scattering
cross sections for BC with mobility diameters below ~350 nm, because of uncertainties associated with theoretical
calculations and laboratory scattering measurements for small particles (He et al., 2015). The increase in BC
scattering during aging was much stronger than absorption, ranging from a factor of 3 to 24 depending on BC size,



morphology, and aging stage (He et al., 2015). Clearly, a proper description of optical properties of particles (along with fuel inventories, emission factors, remote observations, etc.) is essential for analyzing and predicting the climate impacts of BB.

These radiative balance calculations require knowledge of aerosol optical properties, including single scattering albedo (SSA), scattering and absorption cross-sections and efficiencies, and angstrom coefficients. SSA, in particular, is crucial for predicting the direct radiative forcing of an aerosol. A number of experimental techniques have been used to measure the optical properties of BB aerosols (Bond et al., 1999;Holben et al., 1998;Arnott et al., 1999;Arnott, 2003;Haywood et al., 2003;Clarke et al., 2004;Petzold and Schonlinner, 2004;Schnaiter et al.,
2005b;Lack et al., 2006;Moosmüller et al., 2009). By combining photoacoustic spectroscopy (PAS) with Nephelometery, one can simultaneously measure of both absorption and scattering (Chakrabarty et al., 2014;Nakayama et al., 2013;Lewis et al., 2008;Gyawali et al., 2012;Flowers et al., 2010;Wang et al., 2014). Massoli et al. (2009) examined the uncertainty in the SSA of absorbing particles, based on measurements that combine cavity ring-down spectroscopy (CRDS) for extinction measurements with either nephelometry for scattering or PAS
for absorption. Uncertainties in SSA using nephelometer data are larger and are most significant for SSA < 0.7 (Massoli et al., 2009). Massoli et al. (2009) observed nephelometer scattering cross section errors using the Anderson and Ogren correction method to be 3 % at SSA = 1, increasing to 5 % at SSA = 0.7, and 29 % at SSA = 0.4 (Fig. 7. in their work). This is the main contributor SSA error, which is 30% at SSA = 0.4. They reduced this to 25 % using an alternative scheme for deriving a correction for the instruments inability to measure high and low
scattering angles ($C(\lambda)$), but the CRDS/PAS combination yielded SSA errors of between <1 % at SSA = 1 and 8 % at SSA = 0.4. (Massoli et al., 2009). This will be a significant source of error for measurement environments of fresh biomass burning plumes. In comparison, they report an uncertainty of <2 % for this same range when photoacoustic absorption measurements are combined with CRDS (Massoli et al., 2009).

A sensitive technique for measurement of SSA is the combination of CRDS to measure extinction (scattering+absorption) and integrating nephelometry for measuring scattering. CRDS promises aerosol extinction measurements with accuracies of 2 % or better (Smith and Atkinson, 2001;Strawa et al., 2003;Pettersson et al., 2004) and the integrating nephelometer has a reported accuracy of ∼ 7 % (Anderson and Ogren, 1998). The extinction and the scattering coefficients are measured simultaneously for the same aerosol sample in this system,
though we are not sampling the same space in this work. This so-called the extinction-minus-scattering technique has been used for in situ measurements of aerosols (Hallar et al., 2006), studying the optical effects of organic coatings on particles from BB (Riziq et al., 2007;Riziq et al., 2008), retrieving complex refractive indices of humic–like aerosols (Dinar et al., 2008), and determining the SSA of isolated aerosol particles (Butler et al., 2007;Miller and Orr-Ewing, 2007). In our recent calibration study (Singh et al., 2014) we have accounted for errors due to
differences in particle concentration between the condensation particle counter (CPC), nephelometer, and CRDS. Instead of using absolute concentration values, the number density ratio between the CRDS and nephelometer, based on loss measurements, was used to derive SSA. We found that this is the only method in which SSA values are at all





useful, which is 1.7–4.3 % (2.1 % average) for particles ≥200 nm in diameter, as opposed to ~14.3 % for other

methods. The run-to-run variability of SSA measurements is ~2 %. At two standard deviations, SSA values of ≤ 0.91 can safely be determined using this technique, with ≤ 0.96 achievable on average. That is, this is the SSA value where CRDS and nephelometer values can be statistically differentiated at two standard deviations.

In this article, we report extinction, scattering, absorption, and SSA measurement results of freshly emitted soot aerosols from burning white oak, red pine, and cedar wood. Our main goal is to obtain a base line (i.e. fresh soot) to

compare these same properties measured as aerosols age. Most current measurements are limited to a single or few discrete wavelengths. The accurate measurement of aerosol optical properties over the *entire solar spectrum* is currently a technological challenge (Ramanathan and Carmichael, 2008). Accurate and realistic interpretation of aerosol radiative properties obtained by remote sensing and space-based measurements requires accurate measurements of the optical properties of aerosols in the laboratory. Featured absorption cross sections need to be

determined, instead of assuming a power law relationship, which requires more effort and advanced instrumentation than single wavelength measurements. We report measurements of optical properties at a wide range of wavelengths to determine absorption cross sections as a function of wavelength which does not rely on any power law relationship**.**

## 2. Experimental methods:

### 2.1. Cavity ring-down setup

Details of the experimental method and derivation of key equations for particle optical properties and CRDS analysis have been described (Singh et al., 2014) and references therein. We provide here the key equation and summary of the experimental procedure. The extinction coefficient $\alpha_{ext}$ (m$^{-1}$) is defined by

$$\alpha_{ext} = \frac{R_L}{c_{air}} \left( \frac{1}{\tau} - \frac{1}{\tau_0} \right) = \sigma_{ext} N_{CRD} \tag{1}$$

Where $c_{air}$ is the speed of light in air and $R_L$ is the ratio of mirror-to-mirror distance $d$ to the length of the cavity occupied by the sample, resulting in a unit less value >1. The exponential decay of light exiting the cavity is characterized by the time needed for the light intensity to drop to $1/e$ of its initial intensity value, which is $\tau_0$ in an empty cavity and $\tau$ in the presence of a sample. Extinction of an individual particle is characterized by its extinction cross section $\sigma_{ext}$ (m$^2$/particle) and is related to the coefficient by the number density of particles $N_{CRD}$

(particles/cm$^3$) in the CRD cavity.

The laser components of the system, shown in Fig.1, included a Continuum Surelite I-20 Nd: YAG laser running at 20 Hz. The 532 nm beam pumped a single grating ND6000 dye laser with a bandwidth of 0.08 cm$^{-1}$ at 560 nm, and the 355 nm beam pumped an optical parametric oscillator (OPO) laser. The OPO laser was also coupled to the ring-

down cavity which allowed a wider wavelength range than can be achieved with the dye laser, though with decreased beam quality. To retain most of the light exiting the OPO, the beam needed to be reshaped with $f = 40$ cm





achromatic lens and an iris. The OPO had a relatively collimated beam ~1 cm in diameter and its bandwidth was ~0.9 nm in the vicinity of 550 nm light, ~2 nm around 600 nm, and ~9 nm at 680 nm. This type of OPO had a bandwidth that increases asymptotically as it approached 710 nm (twice the 355 nm pump), so the dye laser was

used in this region and the OPO was used at shorter wavelengths. A polarizer and $\lambda/4$ wave plate were used to isolate the lasers and a telescope was used to mode match the laser with the cavity. Since the mirrors were reflective over a limited wavelength range, several sets of mirrors were used to cover a wide wavelength range.

The CRDS system was controlled by a combination of commercial (Continuum) and home-built software. The ring-

down measurements were recorded and analyzed in LabVIEW (National Instruments, version 8.6). The exponential decay was plotted in a log format and a line was fit between two cursors to determine the slope and, therefore, $\tau$. The laser wavelength was also controlled through LabVIEW, where the calibration of the laser wavelength was handled in LabVIEW for the dye laser and internally in the Continuum control software for the OPO. The dye laser was calibrated against a wave meter (Bristol Instruments, model 821B-Vis) over a range of wavelengths. WCPC

measurements used Aerosol Instrument Manager (TSI) and the nephelometer used NephWin (TSI) software

### 2.2. Aerosol generation system

The aerosol processing and CRDS setup was similar to the one described by Rudich et al. (Spindler, 2007) with the only difference being the use of a single CPC and the use of both an OPO and a dye laser as light sources. A coupled differential mobility analyzer (DMA)-CPC (i.e. a scanning mobility particle sizer (SMPS)) was used to determine

the size distribution of aerosols. The current experimental setup is described below and shown in Fig. 2. A constant output nebulizer (TSI, model 3076, modified) in recirculation mode was used to generate aerosols from an aqueous solution of suspended particles. This nebulizer was operated by supplying 35 psi of filtered $N_2$. This was fed into diffusion drier (TSI, model 3062) to remove most of the water. The flow from the nebulizer was quite high (3.2 sL/min), which necessitated splitting before the particle flow entered the DMA. Flow from the nebulizer entered the

710 μm impactor inlet, neutralizer, and long DMA that contained a model 3080 (TSI) electrostatic classifier, where the aerosol was size selected. Flow through the entire system (0.58 sL/min) was produced by a pump within the CPC and the DMA sheath flow as 6.0 L/min in single blower mode. Aerosol flow then entered a ring-down cavity (170 cm long, stainless steel, ½" OD), where the aerosol extinction was measured over a range of wavelengths. Aerosol scattering coefficients were then measured at 453, 554, and 698 nm using the integrating nephelometer (TSI, model

3563), and particle concentration was measured by the water-based CPC (WCPC, TSI, model 3788). A purge flow was applied to custom mirror mounts (NOAA-ESRL) at both ends of the CRD cavity to maintain mirror reflectivity.

This gas passed from the $N_2$ cylinder to a mass flow controller (MFC, Sierra Instruments, 20 mL/min) and was cleaned using an inline HEPA filter (TSI, model 1602051) before the flow was split evenly between purge mirrors.

Aerosols were passed through stainless steel fittings (Swagelok) and conductive graphite-impregnated silicone (TSI). All tubing and instruments were connected physically and electrically, and placed at ground potential to maximize the transmission of charged aerosols though the system. All flows, except the DMA sheath flow, were



calibrated against a NIST-certified flow meter (Mesa Laboratories, model Definer 220) that was factory calibrated annually and had a listed accuracy of <1 %.

### 2.3. Burning facility

Soot was generated with a burning drum designed in our laboratory (Fig.3) and burns were conducted at an off-campus location. This burn drum was equipped with adjustable vents and a lid that was attached to a support structure so that it could fit tightly over the drum or only partially cover it, as needed. Smoke moving through the lid exits a steel chimney pipe, which was sampled by a ½" ID copper tube that acted as a passively-cooled heat exchanger. A Teflon tube connected the heat exchanger to a cross, where particles are sampled by a cascade impactor, a liquid impinger (AGI-4) (Giordano et al., 2014), and SMPS. This resulted in a suspension of black carbon in water, though some of it may have dissolved (Miljevic et al., 2012). A sampling time of $\leq$ 30 minutes, a volume of collection fluid of 30 mL, and a flow rate of 12.5 L•min$^{-1}$ was used (Reid et al., 2013). The residence time from combustion to sampling was on the order of tens of seconds. All wood samples consisted of heartwood, sapwood, cambium, and bark. They were not green, and were air dried for at least several months prior to burning. A Sioutas cascade impactor was used with a Leleand Legacy pump to collect the soot on aluminum filters, to allow visual analysis using a scanning electron microscope (SEM). The samples were collected between 30 seconds and 3 minutes, depending on aerosol load. For times much longer than this, the pump on the impactor began to clog and our filters became saturated. The filters were then removed from the impactor and stored in Ziploc bags until later analysis. Filters were analyzed without further processing on a Zeiss EVO SL10 SEM.

The impinger, which contains water, is transferred to glass bottles with Teflon-lined lids and brought to our lab. Samples were diluted and sonicated prior to introduction to CRDS and nephelometer, and samples were agitated using a magnetic stir bar throughout atomization. The TSI atomizer had been modified to incorporate a stir plate and accept wide-mouth bottles, which reduce the number of sample transfers and decrease the likelihood of sample carryover. Samples were characterized for their particle size distribution before and after nebulization using an SMPS, and several sizes of soot particles were selected for measuring their optical properties. Baseline measurements were taken with nebulized water without particles to take into account any possible particles generated from residues in the water and to minimize the change in water vapor concentration between the blank and particle measurement experiments. The DMA was set to the same pass diameter as the normal particle measurement, and $\tau_0$ was recorded at either 0.2 or 0.5 nm increments over the wavelength range of the mirror. At each wavelength, $\tau_0$ was measured at 20 Hz for 30 seconds, producing an average of 600 measurements. Values were averaged over 3 separate experiments. Blank scattering coefficient measurements were recorded by the nephelometer during this time period and, though they were several orders of magnitude smaller than particle measurements, were subtracted from particle measurements. Particles were then introduced to the system and readings from the CRDS, nephelometer, and WCPC were allowed to stabilize.





The extinction and scattering cross section of fresh soot from white pine, red oak, and cedar were measured using the CRDS and nephelometer, which were used to calculate the absorption cross section and single scattering albedo. The measurements were made for two wavelength ranges 500–580 nm and 580–660 nm. Earlier the measurements were done on different days for each wavelength range, since it involved changing mirrors and conducting realignment of the laser beam, though some later measurements were performed on the same day. Particle number density also varied somewhat over the experiment, but remained mostly consistent. Experiments where the number densities were found to fluctuate significantly were disregarded. The ring–down time was different in each run due to different HR mirrors used for each wavelength range. The error was higher on the low end of the spectrum, due to decreased mirror reflectivity. We have extensively discussed the method used to calculate optical parameters and their associated errors in Singh (2014), and the same method is applied to soot samples in this work.

## 3. Results and discussion

The initial sample collection was done on Nov. $3^{rd}$, 2014. The method of soot collection in distilled water has not been previously reported. We cannot account for any chemical modification of the soot during impingement with our current instrumental capabilities. A comparison of the size distributions of white pine during combustion and after renebulization showed a change in the particle size distribution. The flaming sample had a mode number-density diameter of 148 nm during the burn (interpolated from peak edges due to detector overload) and 55 nm upon nebulization. The smoldering stage sample similarly went from a mode diameter of 138 nm during the burn to 50 nm after nebulization. In both cases, the mode diameter is reduced by a factor of ~2.7. However, it is not known if the optical properties of size-selected particles change due to this sampling process. Once soot was in solution, though, it *did not display any change in size distribution* over several weeks, which makes it a potentially viable method of sample storage. Chemical analysis would be needed to account for any chemical change, which was not available in this work. Future work will explicitly investigate how well this sampling method conserves the optical properties of size-selected soot. Since some aerosol properties change following impingement and nebulization, the state of soot in this work could be more reflective of fresh soot that has undergone cloud processing in the atmosphere (i.e. deliquescence followed by droplet evaporation). However, many important processes did not take place for these samples, including photochemical transformations. As such, results in this work will only be compared to literature observations of fresh soot.

Extinction, scattering, and absorption cross sections showed a decrease at higher wavelengths for measurements done several weeks apart (i.e. a different wavelength range at a different time). Even at overlapping wavelengths, older samples had lower cross section values, resulting in an abrupt discontinuity at 580 nm (the boundary between the ranges of the two sets of mirrors used in this work). Even though the size distribution did not change over the course of several weeks, we attempted to attribute the decrease in optical values to either changes in chemical properties of the soot or to an experimental artifact. When measurements were done on the same day for both wavelength ranges, the abrupt change in the measured values was reduced for most runs; showing this discontinuity. The discontinuity could be a result of several factors including extinction coefficient error is about 1.3–1.7 % (1s)





and the run-to-run variability is similar (~2 %); variation in the particle concentration between runs, so the cross
section error is just over 10 %. And it could be due to the actual changes in the sample. To adjust measurements
performed on different days, a constant was derived from the difference in cross section values at 580 nm for each
wavelength range. While values from 500–580 nm were kept the same, this constant was added or subtracted to
cross section values 580–660 nm. In general, the cross sections of soot particles decreased with increasing
wavelength.

Figure 4 shows the steps followed in determining cross sections, SSA, and their errors. For extinction, the
coefficient is measured at a particular size and wavelength multiple times (individually denoted by *). The error
(one standard deviation, $s$) is derived from this. The relative standard deviations ($RSD$) of several factors are used to
calculate the average cross section and average error from the original extinction coefficient. A similar process is
shown for scattering with the inclusion of a correction factor and its associated error. A broadband correction factor
$A_{Neph}$ is used to reconcile scattering with extinction for completely scattering particles (Singh, 2014). The empirical
Ångström exponent-based correction of Anderson and Ogren (1998) was used to account for truncation angle error.
The SSA and its error are based on scattering and extinction coefficients, the $RSD$ of those coefficients, and the
relative number density in the CRDS and nephelometer. The error for each quantity is calculated using Eq. (2) in
(Singh et al., 2014). The error of extinction coefficient is calculated from Eq.(2).

$$s(\sigma_{ext}) = \sigma_{ext}\sqrt{\left(\frac{s(N_{CPC})}{N_{CPC}}\right)^2 + \frac{1}{N_{RSD}Vrt_s} + \left(\frac{s(R_L)}{R_L}\right)^2 + \frac{s(\tau_o)^2}{(\tau - \tau_o)^2}\left(\frac{\tau_o^2}{\tau^2} + \frac{\tau^2}{\tau_o^2}\right)} \qquad (2)$$

Where $s(\sigma_{ext})$ is the standard deviation in the extinction coefficient, $N_{CPC}$ and $s(N_{CPC})$ are the number density in
measured by the CPC and its error, respectively, $N_{RSD}$ is the number density in the ring-down cavity, $R_L$ is distance
occupied by the sample relative to the mirror-to-mirror distance, $s(R_L)$ is the error associated with measurement of
$R_L$, $t_s$ is the averaging time (30 s), $r$ is the sampling repetition rate (20 Hz), and $V$ is the beam volume. $s(\tau_o)$ is the
standard deviation of the blank ring–down time.

The particle number density in the cavity ($N_{CRD}$) is assumed to be, on average, between the particle concentration
entering and exiting the CRDS. By measuring the particle loss in nephelometer ($L_{Neph}$) and cavity ($L_{CRD}$) for each
particle size, the number density in the nephelometer ($N_{Neph}$) and cavity is calculated from the CPC measurement
($N_{CPC}$) using the following equation

$$N_{Neph} = \frac{N_{CPC}}{L_{Neph}} \quad \text{and} \quad N_{CRD} = \frac{N_{CPC}}{2L_{Neph}}\left(\frac{1}{L_{CRD}} + 1\right) \qquad (3)$$

The accuracy of SSA based on extinction and scattering, is limited largely by the nephelometer at low SSA values.
Specifically, the truncation angle correction $C(\lambda)$ is the limiting factor, which is discussed by Bond (2009). Changes
in the particle size alter the degree of angular scattering, which in turn changes $C(\lambda)$. $C(\lambda)$ is expected to have a $\leq 1$
% error for SSA values greater than 0.9, < 2 % for values 0.8−0.9, and ~5 % for values lower than 0.7. Bond et al.





suggest that, if Anderson & Ogren-corrected SSA is larger than 0.9, then using $C(\lambda)$ is acceptable. Under other conditions where the error is considered unacceptable, especially those found in laboratory or field measurements of biomass or biofuel combustion, the $C(\lambda)$ systematic error can be as high as 5 %. To reduce the scattering error for

low SSA particles, it is suggested that the size distribution should be measured and a refractive index should be assumed. Assuming a refractive index does not cause systematic errors above 2 %, but this method is definitely difficult and size distribution errors must be constrained to avoid error commensurate with using the Anderson & Ogren method. Assuming this error can be maintained ≤2 %, a mean SSA error of ≤2.9 % is expected using this method (Singh et al., 2014). Using the method of Bond et al. (2009), scattering error was estimated using the

observed SSA and Ångstrom absorption exponent (AAE), and was found to be 3–6 %. The uncertainty for using CRDS and nephelometry with this technique to measure SSA of fresh soot is estimated to be less than 2–6 %.

As BB particles age, aerosol growth is not the only means in which they change. Often, dilutors are used in laboratory and field experiments on BB emissions to represent dilution due to diffusion in the atmosphere. These

coatings can evaporate substantially during dilution of a smoke plume to ambient conditions. While the generation of volatile compounds cannot be ruled out in our work, we did not take into account the impact this may have on the optical properties of the soot samples collected in this work. In the sampling system used in this work, any coating on the soot could be lost (i.e. dissolved) after being impinged and would make the measurement of the re-suspended soot core drastically different from a core-shell or more complex coating structure that might be generated.

Alternatively, previously uncoated particles could be coated with water soluble, but non-volatile or semi-volatile species. We aim to systematically address these issues in future work, when these measurements become available.

It has been shown that the presence of large, multiply charged particles passed by the DMA can artificially increase measured cross sections, even if their number density is relatively small. An inline impactor with a 1 μm cutoff diameter or larger has been successfully used to exclude multiply charged particles from a gas stream (Mellon et al.,

2011). Of course, this method is limited to particle diameters of 500 nm or larger. For smaller particle diameters, a separate experiment must be performed. NIST scientists have used aerosol particle mass analyzer (APM), to take an aerosol stream that has been size selected with a DMA and separate it by mass. Consequently particles that have the same electrical mobility, but different mobility diameters were separated. This method had its own limitations for irregularly shaped particles (in this case, it includes soot) (Radney et al., 2013). Unfortunately, our lab is not

equipped with this instrument and possible errors due to multiply charged species have been ignored. Previous work in our laboratory on polystyrene spheres revealed that multiple charging and surfactant coating could significantly increase extinction and scattering measurements when these were compared to Mie theory (Singh et al., 2014). It was found that the technique used in this work, and similar instruments, was limited to particles with diameters ≥200 nm, which was a restriction followed in this work.

The SSA as a function of wavelength for fresh soot produced from cedar, red-oak, and white pine had a slope close to zero over the wavelength range of 500–680 nm, with values ranging from 0.46 to 0.74. While our measured optical properties of fresh soot are within the range of values measured by other groups, reflecting both the dynamic



nature of fires, these variations may be due to significant differences in smoke aging processes, burning conditions,
sample handling and processing, and measurement techniques used (Schnaiter et al., 2005a;Lewis et al., 2008;Mack
et al., 2010;Liu et al., 2014). The SSA for cedar, red-oak, and white pine were plotted for 300, 400, and 500 nm
particles during the flaming stage (Fig. 5, 6 and 7) and during the smoldering stage (Fig. 8, 9 and 10). The solid lines
represent the mean values of SSA and the dotted lines represent error about the mean. The mean values and their
errors are shown in Table 1. Cedar had an SSA that was significantly greater for the flaming stage than the
smoldering stage for 300 nm particles. The smoldering stage was greater than flaming for larger particles, but not
significantly for 400 nm particles. Red oak had SSA values that were comparable for 300 nm particles, greater for
smoldering for 400 nm particles, and slightly greater for smoldering 500 nm particles. White pine behaved similarly,
though the flaming stage had a slightly greater SSA for 300 nm particles. The smoldering combustion phase has
been observed to emit larger particles with a higher scattering efficiency (Chen et al., 2006). Smoldering fires often
lead to BrC, which is less absorbing than BC (Chakrabarty et al., 2010). While this may explain some of these
observations, it is clearly not a rule, given contradictory values for 300 nm particles. Additionally, for larger
particles where the mean SSA for smoldering particles were greater than flaming particles, half of them did not have
a statistically significant difference.

The SSA of fresh soot from the smoldering stage for 300, 400 and 500 nm particle sizes, are slightly dependent on
size parameter ($\chi$) and range from 0.46–0.71, as shown in Fig. 11. The SSA of these fuels versus $\chi$ was plotted in
Fig. 12 for the flaming stage, and had values 0.50-0.71. Size-segregated measurements of SSA seem to be more
variable for the smoldering stage than the flaming stage, though this conclusion is based on a limited set of data. For
each fuel investigated, the SSA values of the smoldering stage became slightly larger as particle size increased (i.e.
they becomes more scattering). Reid et al. has suggested that smoldering combustion may produce larger particles
than flaming combustion due to a greater contribution of a non-absorbing component containing OC (Reid et al.,
2005a). This is consistent with the result of others, where flaming-dominated fires had higher mass fractions of BC,
while smoldering fires produced roughly four times as much OC as flaming-dominated fires (McMeeking, 2008).
This is also consistent with Tumolva et al. (2010), who observed that the flaming stage of white oak produced
significant quantities of fractal-like particles, while smoldering pine bark predominantly produced tar ball-like
spheres. For the flaming stage, there was no observable trend as a function of particle size. All species tended to
have equivalent SSA values for 300 nm particles, but diverged significantly at larger diameters. While wavelength
does not seem to significantly affect SSA in this work, the particle size clearly plays major role in determining the
scattering or absorption properties of the particle.

A number of ambient field studies on optical properties of BB aerosols have been done, several of which are
reported in Table 2. These were mainly measured at a single wavelength but not all were done at the same
wavelength. In general, soot particles generated by burning propane or ethylene in the laboratory or emitted from
diesel engines have a much lower SSA than BB soot (Wei et al., 2013;Khalizov et al., 2009a;Schnaiter et al.,
2005a;Schnaiter et al., 2006;Radney et al., 2014). Liu et al. (2014) measured SSA and AAE of fresh BB aerosols





produced from 92 controlled laboratory combustion experiments of 20 different woods (Ponderosa Pine (PP), red oak, wheat straw, rice straw and others) and a relatively fresh plume during a field-based measurement of the Las Conchas wildfire in 2011. They demonstrated that an SSA of BB aerosol spans a large range (~0.2–1) and SSA varies strongly with fire-integrated modified combustion efficiency ($MCE_{FI}$), which is a measure of how cleanly a
fuel is combusted. They found that SSA is close to 1 between 532 and 781 nm, as long as $MCE_{FI}$ is below ~0.85. At higher $MCE_{FI}$ values, SSA drops precipitously and exhibits greater spectral dependence, which corresponds to a lower OC content. This study also showed that both SSA and AAE increase with aging. We find that our SSA values for red oak (0.53-0.68) are fairly close to the observations of Liu et al. (0.45–0.59 at 532 nm). It is possible that the slightly smaller value observed by Liu et al. is due to the use of a cooking stove, which combusts more
cleanly than an open burn. Our flaming and smoldering SSA values for white pine (0.46–0.74) were significantly different than PP; either a mix of brown and green (0.97 at 532 nm) or all green (0.93–0.99 at 532 nm). This was an open burn, performed similarly to our work, which shows that, even within types of pine, drastic differences in SSA (0.19–0.53) can be observed between species.

Bergstrom et al. (2003) performed broadband SSA estimates of the total aerosol column using solar radiative flux and optical depth measurements over 2 days during the SAFARI 2000 field experiment in southern Africa. A detailed radiative transfer model resulted in SSA values from 0.85 to 0.90 at 350 nm, decreasing to 0.6 in the near infrared (Bergstrom et al., 2003). Observations with small optical depth over the ocean showed a slightly decreasing SSA with wavelength; 0.84±0.06 at 500 nm and 0.79±0.11 at 660 nm. When sampling springtime BB haze over
Mongu, Zambia, however, a very high optical depth was observed and SSA had little spectral dependence in this region (0.87±0.01 at 500 nm and 0.86±0.02 at 660 nm). While this SSA is higher than our measurements, the same lack of spectral dependence is observed. In aircraft measurements by Johnson (2008), the SSA of BB aerosols over western Africa during the Dust and Biomass Experiment (DABEX) varied from 0.73 to 0.93 at 550 nm. After removing a contribution for mineral dust, they found an SSA around 0.81±0.08 for both aged and fresh smoke
plumes from agricultural fires. This SSA value is higher than our observations for fresh soot by approximately 0.05– 0.35, which could be due to the presence of silica from the agricultural refuse of many silica-rich crops (millet, maize, sorghum, and other grains) or mineral dust. A similar explanation can be given for differences between our results SAFARI campaign estimates, though no source attribution was performed in this case. Alados-Arboledas et al. (2011) monitored a fresh BB plume using a combination of Raman lidar and star–and sun–photometers, finding
relatively low SSA values of 0.76–0.86, with lower values for fresh BB aerosols than aged smoke. Three wavelength measurements by Liu et al. (2014) shows fire-integrated, fitted SSA values at 405 nm to be slightly smaller than those at 532 nm by 4–5 %, where the difference becomes larger for increasing $MCE_{FI}$, but only slightly. SSA values at 532 and 781 nm are nearly equivalent (5 % difference) over a wide range of $MCE_{FI}$, but begins to diverge drastically when $MCE_{FI} > 0.92$, with the SSA at 532 nm being the larger value. This observation bounds the $MCE_{FI}$
of burns performed in this work to <0.92. While field measurements and remote sensing retrievals of SSA rarely falls below 0.6 in ambient plumes, the differences in SSA among BB aerosols is attributed changes in combustion conditions produced by different fuel types as well as soot age (Eck, 2003;Lewis et al., 2008;Mack, 2010). In



comparison to these field observations, our results are lower than is typically seen for fresh soot, even when external mixing is taken into account. This could suggest that the MCE values of wildfires are higher than controlled

laboratory burns. However, our lowest observed SSA was during a smoldering burn of cedar (0.46 average), which should have a relatively low MCE. The difference in SSA due to fuel type is at least 0.15 and could be much greater (~0.4), when comparing this work to field observations of fresh soot. Clearly, fuel type or some other factor, such as the presence of very scattering particles that are larger than those studied here, play an important role.

Lewis (2008) found SSA values at 405 nm ranging from 0.37 to 0.95 for flowering shrubs and pine needle litter during Fire Laboratory at Missoula Experiment. Chemical and physical properties determined from X-ray and electron microscopy methods found that the combustion products of pine needles, wood, and litter (duff) are chemically similar and their particles consist of liquid oily OC with BC inclusions (Hopkins et al., 2007). PP needles/twigs and duff were found to have a fire-integrated SSA of 0.91 and 0.97, respectively, which is

significantly larger than any of our white pine measurements (Table 1), despite having similar burning conditions. BB aerosols of Southern Longleaf pine needles were also significantly greater, having an SSA of 0.89. While some of this variability can be attributed to $sp^2$ hybridization, which should be related to MCE, SSA values observed in this work were different than wood species investigated by Hopkins et al. SSA values observed in this work were commensurate with shrubs in Hopkins et al., where particles were mainly BC with inorganic inclusions. Given the

presence of tar balls in this work, the variability in the SSA of BrC-coated BC is clearly greater than previous measurements. In the NASA measurements during SEAC4RS, involving in situ sampling of the smoke from the Yosemite Rim Fire, the initial SSA of smoke was 0.92 and increased in the first 0–7 hrs. to 0.96, and was nearly constant after that (up to two days) (Beyersdorf, 2013). This is significantly higher than our SSA values for fuels studied in this work. This area of Yosemite was in the Lower Montane forest zone, which predominantly had

California black oak, PP, incense-cedar, and white fir. Fresh smoke in the Rim Fire is commensurate in SSA to BB particles from PP needles and twigs (SSA of 0.91 at 532 nm) (Hopkins et al., 2007). However, this was lower than either PP duff or a mix of brown and green PP wood, which have an SSA of 0.97 at 532 nm (Hopkins et al., 2007; Liu et al., 2014). This could be due to the presence of significant quantities of very old fuel on the ground, which could produce more efficient combustion. Chen et al. (2006), reported SSA values of 0.35–0.70 for dried PP wood

in controlled laboratory combustion studies, which supports this assertion. Chen et al. (2006) also observed an SSA of 0.70 for white pine needles during the smoldering stage in controlled laboratory combustion studies, which is in fair agreement with white pine soot in our work (0.46–0.74).

AAE values determined in this work are presented in Table 1. By making a $\log_{10}$-$\log_{10}$ plot of the absorption cross

section ($\sigma_{abs}$) vs. $\lambda$, a linear fit was performed to yield the AAE. Significant errors were observed in the range of 580–660 nm due to the mirrors being less reflective. As such, fits were only performed from 500-580 nm. Saleh et al. (2013) measured AAE for a variety of fuels, and found values of 1.38 for fresh oak, 1.42 for aged oak, 1.48 for fresh pocosin pine, 1.73 for aged pocosin pine, and 2.15 for fresh gallberry. These values are consistent with other measurements for BB emissions (Liu et al., 2014; Saleh et al., 2013; Gyawali et al., 2009; Habib et al., 2008). Liu et



al. (2014) observed AAE values for red oak (1.16–1.24) that were lower than our observations for either burning stage (2.13–3.58 for flaming and 3.04–5.57 for smoldering). This difference could be due to the burner, where a cleaner burning flame leads to higher MCE, lower SSA, and lower AAE as the fraction of BC increases. In contrast to SSA, AAE values for white pine in this work (2.17–4.20) are similar to values of Liu et al. for PP (2.9 for mixed brown/green and 1.99–4.60 for green). While they do observe a regime where there are such large values, and large

variability in those values, they observe this for SSA values >0.8 at 405 nm or >0.85 at 532 nm. This is inconsistent with the relatively low SSA values for white pine observed in this work. Cedar is also incongruent with the observations of Liu et al., since relatively large AAE values were observed for the flaming stage (0.70–3.3), but small SSA values (0.50–0.61). The incongruence is even more pronounced for the smoldering stage, where AAE is even higher (1.3–4.7) while SSA is about the same, but more variable (0.45–0.64). Lastly, a lack of wavelength

dependence in SSA was only observed when MCE was <0.92, where SSA values were not observed below 0.8. This suggests a potential issue with the framework developed by Liu et al. For each fuel and particle size, a larger AAE was found for smoldering combustion, compared to the flaming combustion stage, which is consistent with a significant absorption by BrC in the visible region (Chen et al., 2006;Chang and Thompson, 2010). AAE values in this work are generally larger than those observed in relatively fresh plumes from the Las Conchas wildfire (2.1±0.5

at 1σ). This may be due to differences in burning conditions or fuel type, where the majority of burning was in the Southern Rocky Mountain Mesic and Dry-mesic Montane mixed conifer ecosystems and Ponderosa Pine woodland (Bird and Menke, 2011). These areas are dominated by Douglas Fir, White Fir, and Ponderosa Pine, though Blue and Mountain Spruce are frequently present along with a number of shrub types.

The optical properties of aerosols are dominated by their chemical composition and physical characteristics, such as size and morphology, which lead to large uncertainties in quantifying how they directly alter the climate system. Freshly emitted BC particles are mostly hydrophobic and externally mixed with other aerosol constituents (Zhang et al., 2008). There is evidence that fires produce BC particles coated with organic matter in a manner that enhances some of their optical properties, specifically short wavelength absorption by "lensing" (Lack et al., 2012), which

alters the results of climate models (Bond and Bergstrom, 2006). We have observed the same tar ball like particles in SEM images, but we did not perform further analysis (Tumolva et al., 2010). Field measurements indicate that, during transport, fresh soot becomes internally mixed with sulfates and organics, leading to an enhancement of light absorption by about 30% (Schwarz et al., 2008). Backman et al. (2010) measured the effect of heating on light scattering and absorption by aerosols at an urban background station in Helsinki. Heating mixed aerosols would

volatilize scattering, low molecular weight organic constituents, producing an increase in light absorption, with SSA reduced to 0.4 after thermodenuding (Backman et al., 2010). Many of the SSA observations in this work, particularly at 300 and 400 nm diameters, are within this range. The aging process can also affect the morphology of soot by collapsing dendritic structures into a more compact or near spherical morphologies. Particles' ability to act as CCN is largely controlled by aerosol size rather than composition (Dusek et al., 2006). Field measurements

suggest that in mixed aerosol populations, particle size is a good predictor of CCN ability. Aerosols particles can take up water, become larger in size than their dry equivalents, and hence, scatter more light. Wet particles also have



different angular scattering properties and refractive indices than their dry counterparts, even at 50% RH. An internal mixture of soot with other aerosol components is significantly more absorptive than the external mixture (Jacobson, 2000). The optical properties of fresh (uncoated) soot are practically independent of relative humidity

(RH), whereas soot internally mixed with sulfuric acid exhibits significant enhancement in light absorption and scattering, increasing with the mass fraction of sulfuric acid coating and relative humidity (Khalizov et al., 2009b). While these factors are recognized as important in affecting the optical properties of particles, they are not currently well constrained in this work.

### 4. Conclusion

Though there were differences in the size distribution between sampling and nebulization, and chemical analysis was not available for this work, samples appeared to be stable over the course of 2–4 weeks. A systematic study is planned to determine the suitability of this sampling technique for storing soot samples. A direct comparison of cooled and diluted soot with suspended and re-aerosolized soot, examined as a function of wavelength and particle size, would be required. Efforts are currently underway in our laboratory to perform such a study. It is not currently

known if the optical properties of size-selected particles are altered by this sampling process. Changes in mixing state and particle morphology are possible, and not currently constrained in this work. It is not presently known if the optical properties of size-selected particles are altered by this sampling process, and changes in mixing state and particle morphology are possible. The effect of large, multiply charged particles is not likely significant in for particle diameters ≥200 nm, which was a restriction followed in this work.

When samples were stored for more than a few weeks, differences in the extinction and scattering cross sections were observed. A statistical framework, previously developed by our group for analyzing polystyrene spheres (Singh, 2014), was applied to soot, and the error in SSA was found to be 3–6 %. This error was dominated by the truncation angle correction factor $C(\lambda)$. To reduce this error, it would be useful to rely on a scheme that does not

depend on the Ångstrom scattering exponent, which does not well represent particle size at low SSA values. Instead, a method for correcting the nephelometer truncation angle error should be devised for submicron soot particles; is they aggregates of glassy carbon spheres, tar balls, or a mixture of both. To perform this correction, particle size must be selected and the size distribution known.

SSA was determined for fresh BB soot using the extinction-minus-scattering method for a range of particle sizes (300–500 nm) and a wide range of wavelengths (500–660 nm), which is wider than previous direct measurements of BB aerosols. This is important, since the accurate measurement of aerosol optical properties over the entire solar spectrum is a technological challenge that must be addressed to quantify the impact of aerosols on climate. The optical properties (extinction, scattering, and absorption cross sections; Ångstrom absorption exponent; and SSA)

were measured for fresh particles produced from burning white pine, red oak, and cedar. The extinction, scattering, and absorption cross sections decreased slightly toward higher wavelengths, producing a nearly uniform value of SSA for each particle size and fuel source. SSA values ranged from 0.46 to 0.74. Results show that SSA is not



uniformly greater for the smoldering stage than the flaming stage. This was especially true for 300 nm particles, but even for larger particles where the mean SSA for the smoldering stage was greater than the flaming stage, half did

not exhibit statistically significant differences.

While SSA exhibited no wavelength dependence in this work, there was particle size dependence. SSA increased with particle diameter for smoldering fires, whereas flaming fires did not exhibit any trend as a function of particle size. This is likely due to changes in the contribution of tar ball like spheres and fractal BC as a function of particle

size. For radiative transfer models, it is inappropriate to assign a uniform SSA to all particle diameters, which are typically measured for the entire size distribution and integrated over both combustion stages.

In a comparison with literature values, white pine had a SSA that was ~0.1 larger than reported values, likely due to the cooking burner employed by Liu et al. (2014). In comparing different types of pine under similar burning

conditions, significant differences in SSA (0.19–0.53) were observed between species. The lack of SSA spectral dependence seen in this work is consistent with BB haze observations, though field observations of fresh soot typically had higher SSA values than those in this work. This lack of spectral dependence is consistent with $MCE_{FI}$ values of <0.92 (Liu et al., 2014). While MCE clearly influences SSA, SSA differences of 0.15-~0.4 or greater can be attributed to fuel type or fuel state for fresh soot. The relatively low SSA values, however, are consistent with

$MCE_{FI}$ values of >0.92.

Despite the low SSA values observed in this work, AAE values were quite high (1.59–5.57). AAE was larger for the smoldering stage than for the flaming stage, which is consistent with the effects of a greater contribution of BrC in smoldering flames. For white pine and cedar, such large values of AAE are only observed when SSA is >0.85 at 532

530    nm, which is inconsistent with our SSA measurements. When also considering issues with low SSA and a lack of SSA spectral dependence, this suggests there are issues with the MCE-based framework of Liu et al.

Biomass burning is a major global phenomenon with an unusually large number of degrees of freedom, which includes morphology, size distribution, mixing state, age, composition, concentration, location, flaming condition,

fuel type, fuel state, humidity, and chemical oxidants. It is practically impossible to account for all sources of uncertainty, but not all degrees of freedom are equally important. The most significant effects on the intrinsic optical properties of fresh BB particles (i.e. morphology and composition) seem to be burning stage, particle size, fuel type, and fuel condition (green, brown, mixed, littler, etc.). While this work investigates key parameters effecting fresh soot, the optical properties of aged particles are also significantly influenced by mixing state, humidity, and

chemical processes.

Future work involves a plan to design and build an indoor chamber that will be connected directly to the output of a furnace, where additional gases of relevant organic compounds (or proxies of semi volatile species) and nitrogen oxides can be added to simulate atmospheric aging of the BB aerosols. This includes isoprene and many



monoterpenes (like α- and β-pinene), common VOC oxidation products, NO, and $NO_2$, at concentrations that reflect the conditions observed during forest fires. The optical properties of BB particles in the chamber will be monitored as a function of composition and age. The relative quantities of EC and OC will be measured on filter samples, and $MCE_{FI}$ will be determined via CO and $CO_2$ measurements. These chamber CO and $CO_2$ measurements will also be important in controlling burning conditions so that they match CO and $CO_2$ measurements observed during

wildfires. Additionally, furnace conditions can be altered so that different burning states can be investigated. Since satellite measurements of SSA cannot distinguish between BB aerosols and other types of aerosols, controlled experiments that reflect natural conditions are needed to better assess the direct contribution of BB to climate forcing.

**Author contribution**: S. Singh run all the experiments, analyzed the data, Marc Fiddler developed the data analysis
software and design of the experiment. Bililign is the PI of the project and supervised the work.

### 5. Acknowledgements

This work is supported by the Department of Defense under grant #W911NF-11-1-0188. We acknowledge the support from the Joint School of Nanoscience and nanoengineering at NCA&T for the use of the imaging facilities. The authors also acknowledge the contribution of Damon Smith in the field sample collection.



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





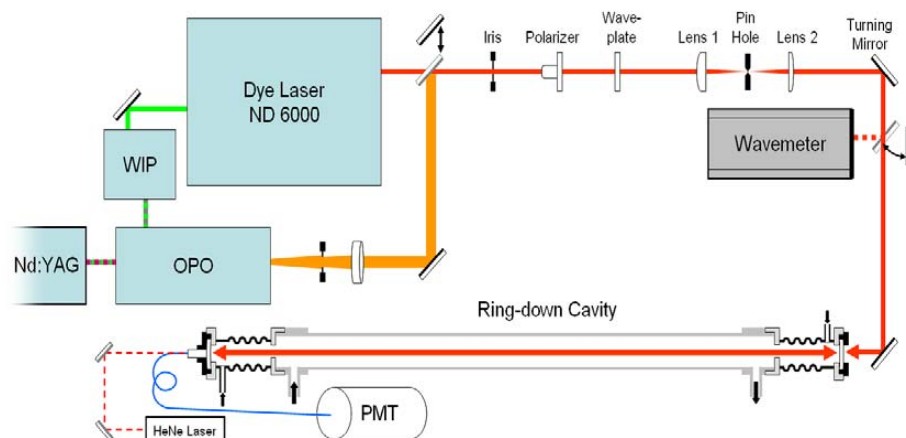

Figure 1.The laser and optical components of the CRDS instrument

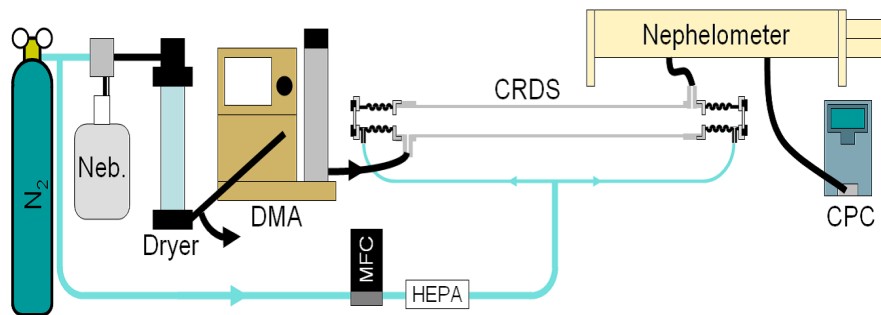

Figure 2.The integrated aerosol optical property measurement system





Figure 3. The soot generation setup, consisting of a burning drum, particle conditioning system, SMPS, and impinger and impactor samplers.

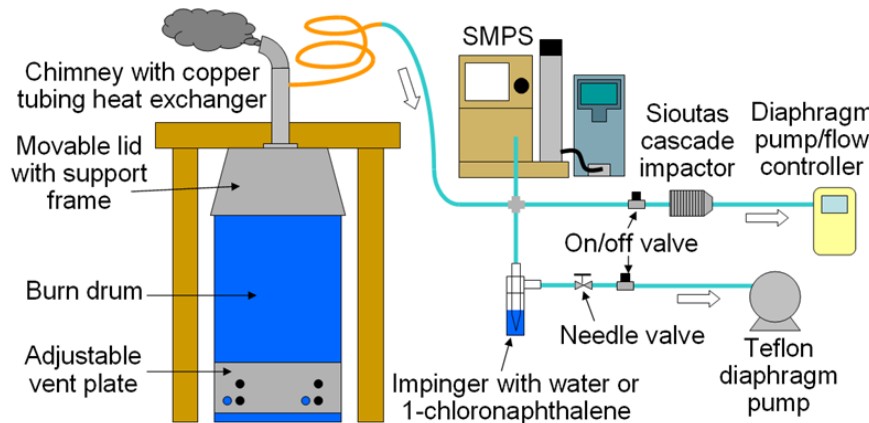

Figure 4. The calculation flow for determining average σext, σscat, ω, and their errors.





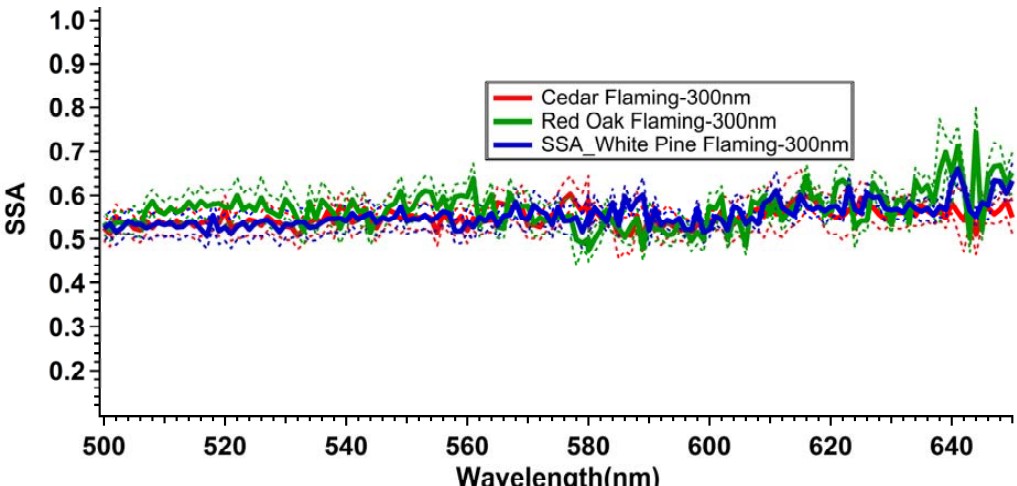

Figure 5. SSA of 300 nm particles from white pine, red oak, and cedar sampled during the flaming stage

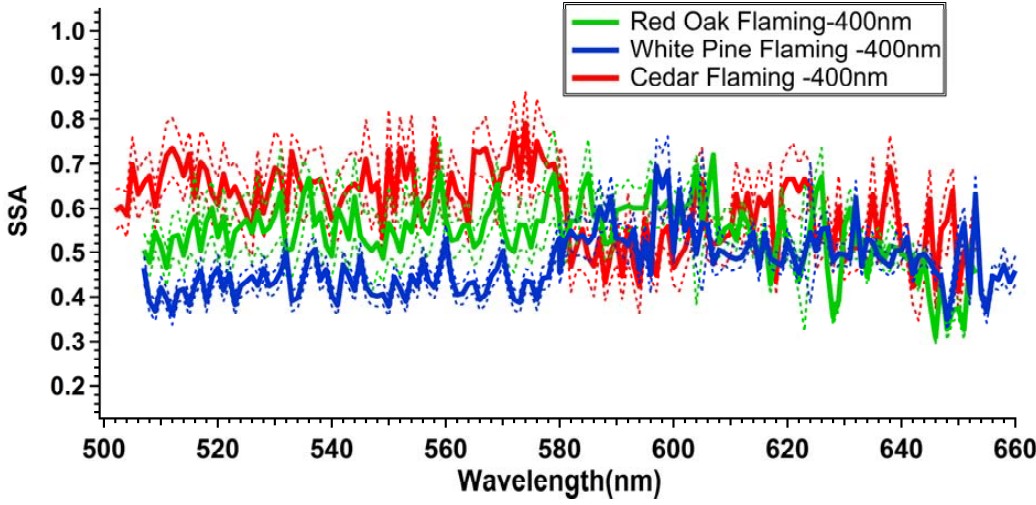

Figure 6. SSA of 400 nm particles from white pine, red oak, and cedar sampled during the flaming stage.

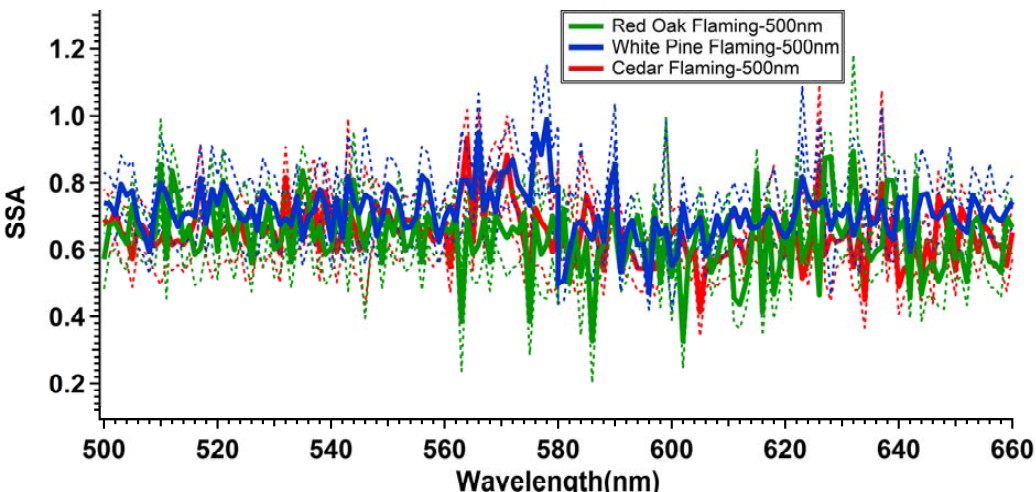

Figure 7. SSA of 500 nm particles from white pine, red oak, and cedar sampled during the flaming stage.

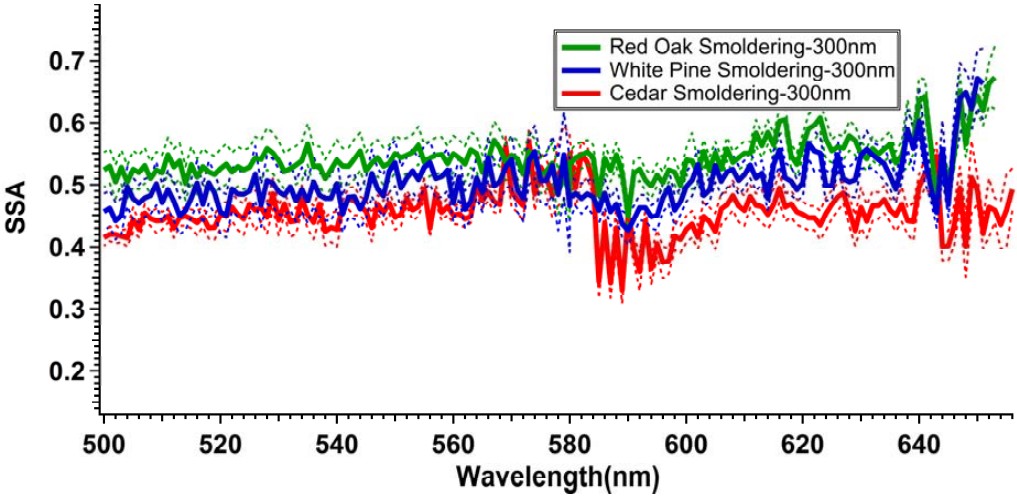

Figure 8. SSA of 300 nm particles from white pine, red oak, and cedar sampled during the smoldering stage.




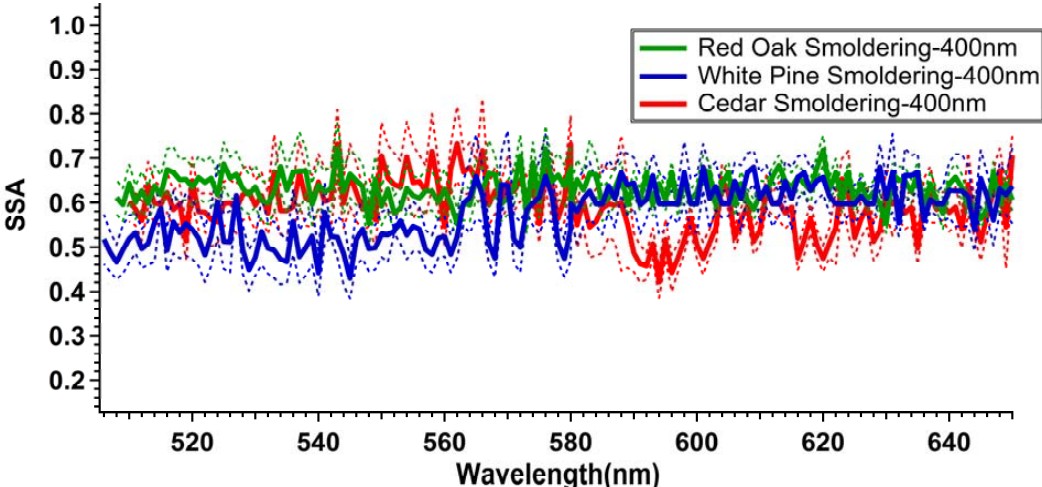

Figure 9. SSA of 400 nm particles from white pine, red oak, and cedar sampled during the smoldering stage.

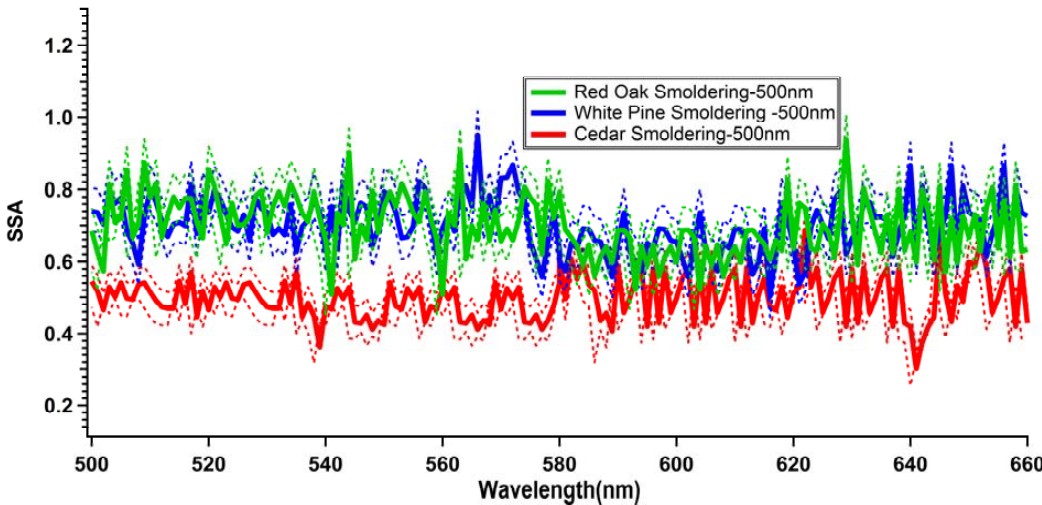

Figure 10. SSA of 500 nm particles from white pine, red oak, and cedar sampled during the smoldering stage.





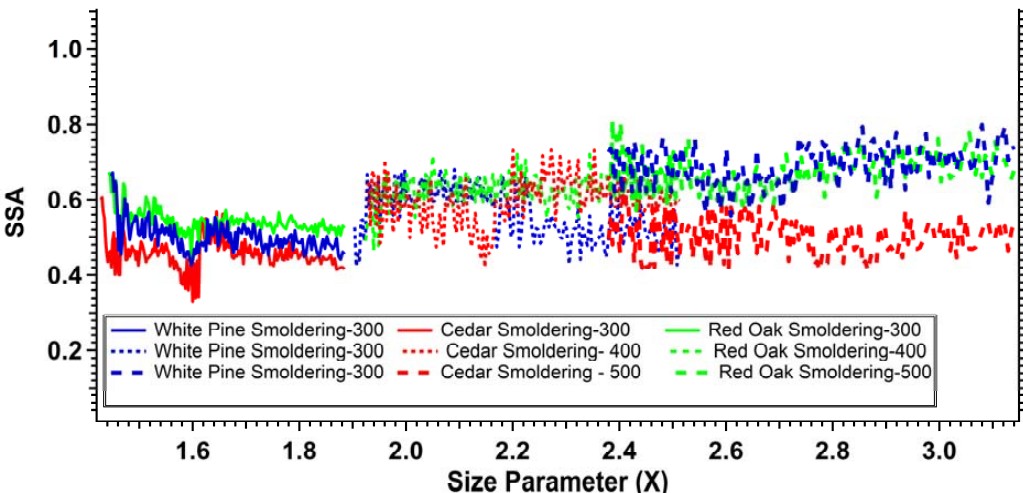

Figure 11. SSA as a function of size parameter for all samples in smoldering stage.

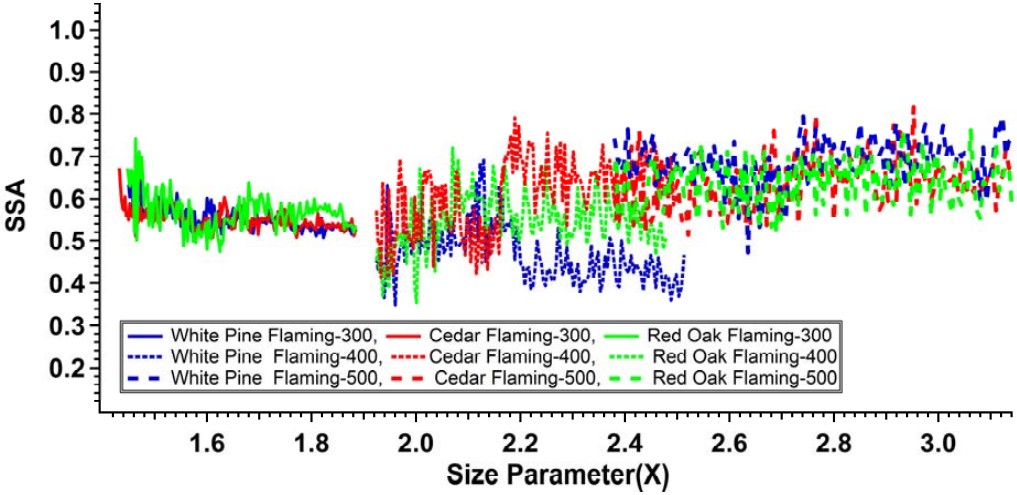

Figure 12 SSA as a function of size parameter for all samples flaming stage.



Table I. Mean SSA values, their error (1σ), and AAE in the 500-580 nm wavelength range.

| Particle Size, Fuel | SSA | | AAE | |
|---|---|---|---|---|
| | **Flaming** | **Smoldering** | **Flaming** | **Smoldering** |
| 300 nm, Cedar | 0.55±0.03 | 0.46±0.03 | 3.34 | 4.53 |
| 300 nm, Red Oak | 0.56±0.03 | 0.54±0.02 | 2.13 | 3.04 |
| 300 nm, White Pine | 0.55±0.05 | 0.50±0.03 | 3.12 | 3.92 |
| 400 nm, Cedar | 0.58±0.05 | 0.61±0.05 | 2.08 | 2.39 |
| 400 nm, Red Oak | 0.53±0.04 | 0.60±0.03 | 3.51 | 3.90 |
| 400 nm, White Pine | 0.46±0.03 | 0.56±0.05 | 3.09 | 4.20 |
| 500 nm, Cedar | 0.50±0.05 | 0.64±0.06 | 1.59 | 2.75 |
| 500 nm, Red Oak | 0.63±0.06 | 0.68±0.06 | 3.58 | 5.57 |
| 500 nm, White Pine | 0.71±0.04 | 0.74±0.06 | 2.17 | 3.24 |

Table II. Previous SSA measurements of fresh BB aerosols

| Reference | Wavelength (nm) | Sample | SSA Range | Method |
|---|---|---|---|---|
| Schnaiter et al., 2005 | 550 | Corn stems | 0.74 | Long path extinction spectrometer and nephelometer |
| Lewis et al., 2008 | 405 and 870 | Laboratory smoke from a variety of biomass fuels, including pine, rice straw | 0.37-0.95 | Dual-wavelength photoacoustic instrument and Nephelometer |
| Mack et al., 2010 | 532 | Laboratory measurements of fresh smoke from wild land fuels in the W and SE US | 0.428-0.99 | Photoacoustic and nephelometer |
| Liu et. al., 2014 | 405, 532, and 781 | Fresh BB aerosols from the controlled laboratory combustion of 20 woods and grasses | 0.2-1.0 | Three-wavelength photoacoustic soot spectrometer and nephelometer |