# Peer review of "Measurement of size dependent single scattering albedo of fresh biomass burning aerosols using the extinction-minus-scattering technique with a combination of cavity ring-down spectroscopy and nephelometry."

_Atmospheric Chemistry and Physics, 2016_

## Short Comment (SC1) · 14 Jun 2016

The paper deals with one the interesting and poorly understood field of biomass burning (BB) aerosol optical properties. I have a couple of questions regarding this study are as follows:

Author's wrote "The SSA and AAE values in this work do not fit well with current schemes that relate these factors to the modified combustion efficiency (MCE) of a

[Figure]

burn" but they have not done proper validation or check because they don't have MCE calculation but present as a major finding in their abstract. They concluded burn condition does not control the SSA and AAE and mention as a one of the major findings in abstract but have not tested SSA and AAE correlation with either MCE or BC/OA ratio. Recently Pokhrel et al. (2016) shows BB SSA shows very strong correlation with EC/(EC+OC). How authors divide flaming and smoldering stages of fires? Page 7 line 249: Author's mentions "even though the size distribution did not change over the course of weeks, but observed decrease in optical values suggests that could be due to changes in chemical properties of the soot, but in line 241 they wrote that result in this work will only be compared to literature observation of fresh shoot". How could chemically change soot be compared with fresh soot? Page 8 line 255: what is the logic behind to adjust 580-660 nm range, not 500-580 nm range? Page 10 line 329: Author's mentions Cedar have higher SSA at flaming state than the smoldering stage for 300 nm particles. What is the region behind this? Do authors want to say organics absorb more than black carbon in 500-660 nm? Since more BC is produced during flaming and more OC is produced during smoldering (Ward et al., 1992). Page 10 line 351-354: Author's mentions there was no observable trend of SSA as a function of particle size and again wrote SSA diverged significantly at larger diameters. What do these sentences signify? And in Line 353 they mention particle size clearly plays the major role in determining the scattering or absorption properties. How that does not affect the SSA? Page 11 Line 394: Author's concluded that due to lack of variability in their SSA values for different wavelength the MCE of the burns for their work is < 0.92 based on Liu et al. (2014) study. If so, then why they called flaming for such burns? Based on Yokelson et al. (1996) definition, MCE of 0.9 represents roughly equal amount of flaming and smoldering and MCE ∼0.8 are pure smoldering. Page 12 Line 398-400: Author's mention that higher SSA values in field measurements than their observation suggest that the MCE values of wildfires are higher than controlled laboratory burns. They did not explain why burn with higher MCE could have higher SSA values. It is clear from laboratory studies that burns with higher MCE have lower

SSA (Pokhrel et al., 2016; Liu et al., 2014; McMeeking et al., 2014) supporting the fact that more BC will produce during flaming stages of burns (Ward et al., 1992). Page 12 Line 408-410: Author's mentions, despite of having similar burn conditions with Hopkins et al. (2007), they found different SSA values for white pine than that of ponderosa pine needles/twigs. But they do not mention how they compare the burning conditions because they don't have either BC/OA or MCE calculations in their study. Throughout the document, authors wrote soot. Do they want to say only soot produced during biomass burning?

References: Pokhrel, R. P., Wagner, N. L., Langridge, J. M., Lack, D. A., Jayarathne, T., Stone, E. A., Stockwell, C. E., Yokelson, R. J. and Murphy, S. M.: Parameterization of Single Scattering Albedo (SSA) and Absorption Angstrom Exponent (AAE) with EC/OC for Aerosol Emissions from Biomass Burning, Atmos. Chem. Phys. Discuss., 16, 1–27, doi:10.5194/acp-2016-184, 2016.

Ward, D. E., Susott, R. a., Kauffman, J. B., Babbitt, R. E., Cummings, D. L., Dias, B., Holben, B. N., Kaufman, Y. J., Rasmussen, R. a. and Setzer, a. W.: Smoke and fire characteristics for cerrado and deforestation burns in Brazil - BASE-B experiment, J. Geophys. Res., 97, 14601–14619, doi:10.1029/92JD01218, 1992.

Liu, S., Aiken, A. C., Arata, C., Dubey, M. K., Stockwell, C. E., Yokelson, R. J., Stone, E. a, Jayarathne, T., Robinson, A. L., Demott, P. J. and Kreidenweis, S. M.: Aerosol single scattering albedo dependence on biomass combustion efficiency: Laboratory and field studies, Geophys. Res. Lett., 41, 742–748, doi:10.1002/2013GL058392, 2014. Yokelson, R. J., Griffith, D. W. T., and Ward, D. E.: Open path Fourier transform infrared studies of large-scale labo- ratory biomass fires, J. Geophys. Res., 101, 21067–21080, doi:10.1029/96JD01800, 1996.

McMeeking, G. R., Fortner, E., Onasch, T. B., Taylor, J. W., Flynn, M., Coe, H. and Kreidenweis, S. M.: Impacts of nonrefractory material on light absorption by aerosols emitted from biomass burning, J. Geophys. Res. Atmos., 119, 12272–12286,

doi:10.1002/2014JD021750, 2014.

Hopkins, R. J., Lewis, K., Desyaterik, Y., Wang, Z., Tivanski, a. V., Arnott, W. P., Laskin, a. and Gilles, M. K.: Correlations between optical, chemical and physical properties of biomass burn aerosols, Geophys. Res. Lett., 34(18), 1–5, doi:10.1029/2007GL030502, 2007.

---

## Author Comment (AC1) · 27 Jun 2016

Responses to questions and comments by Rudra Pokhrel

COMMENT: Authors wrote "The SSA and AAE values in this work do not fit well with current schemes that relate these factors to the modified combustion efficiency (MCE) of a burn" but they have not done proper validation or check because they don't have MCE calculation but present as a major finding in their abstract. They concluded burn

condition does not control the SSA and AAE and mention as a one of the major findings in abstract but have not tested SSA and AAE correlation with either MCE or BC/OA ratio. Recently Pokhrel et al. (2016) shows BB SSA shows very strong correlation with EC/(EC+OC).

AUTHORS RESPONSE-The authors did not have the capability to measure CO and CO2 to derive MCE at the time of this work, nor was MCE recognized is being important until much later. We also lack the capability to measure EC and OC content. We are currently working to address these important aspects of BB aerosols, including chemical analysis and optical properties as a function of aging using an indoor smog chamber in forthcoming work. We thank Pokhrel for bringing his work to our attention. However, the authors did not conclude that combustion conditions and MCE have no effect on SSA or AAE. We recognize that these factors (including fuel type, fuel state, and burning process) are important aspects of BB. The focus of this work was on fuel type. The discussion of the conclusions in question can be found on lines 439 through 446 on page 13. In short, such large AAE values were only seen in the scheme of Liu et al. at SSA values >0.85 at 532 nm, though much smaller SSA values were observed in our work. A lack of SSA wavelength dependence was only observed below an MCE of 0.92, thought the absolute values of SSA were very low and correspond to MCE values greater than 0.92; it cannot be both. The lack of spectral dependence in SSA was based on the fitting parameters found in Liu et al. It appears that Pokhrel et al. arrived at different coefficients, though it is not clear why "allowing the coefficients to vary" would produce this result. The authors will determine if this argument is still consistent with the coefficients found in Pokhrel et al.

For white pine, we observed AAE values from 2.17-4.20. This would correspond to a EC/(EC+OC) less than 0.2, as shown in Figure 4 of Pokhrel et al. However, Pokhrel et al. show SSA values greater than ~0.75 at 532 nm for EC/(EC+OC) less than 0.2. Our observed SSA values range from 0.46-0.74, which is only seen at EC/(EC+OC) significantly greater than 0.2. The flaming stage of cedar does not seem to be as

Interactive
comment

problematic in EC/(EC+OC)-based scheme. These conclusions will be included in the final paper.

COMMENT: How authors divide flaming and smoldering stages of fires?

AUTHORS RESPONSE-This differentiation was done visually. The visual differentiation was based Tillman's description of combustion of wood as three distinct, but overlapping stages (Tillman 1981). The first ignition stage involves use of heat to drive off moisture and bring the wood to the pyrolysis temperature. In the second, stage (flaming), the wood undergoes pyrolysis (thermal decomposition under oxygen-poor conditions) when it reaches around 500-600°F. This process leads to production of organic gases with increasing high molecular weight as combustion progresses. The remaining portion is charcoal, which burns at about 1100°F. Once volatiles are driven off, direct combustion of black carbon occurs (smoldering). Any unburned gas-phase residue will be in the form of smoke or condensed pyrolysis gases. Complete combustion requires plenty of oxygen and the three elements of temperature, turbulence, and time.

Tillman, D.A., Rossi, A.J., Kitto, W.D., (1981). Wood Combustion: Principles, Processes, and Economics. Academic Press: New York, NY.

COMMENT: Page 7 line 249: Author's mentions "even though the size distribution did not change over the course of weeks, but observed decrease in optical values suggests that could be due to changes in chemical properties of the soot, but in line 241 they wrote that result in this work will only be compared to literature observation of fresh shoot". How could chemically change soot be compared with fresh soot?

AUTHORS RESPONSE-The authors recognize that some chemical changes had occurred over the course of several weeks, and took measures to account for these changes. For the most part, these changes seem quite slow. The samples were kept in distilled water and not exposed to the environment. We have several reasons for comparing our values to fresh BB soot. While some changes in the particle size distribution were observed upon impingement, this sampling scheme (or, really, any offline sampling scheme) is unlike any natural atmospheric processing. No photochemical changes were allowed to take place. The system would be too dilute for SOA formation via acid chemistry. Semi-volatile species would be almost immediately put into the condensed phase, though solubility would be a more important factor in determining their interactions with the particulate phase. The consistency of the measurements (with very small changes) done following sample collecting and several days and weeks later also gives us confidence that the chemical change is not significant. In short, our samples had more in common with the physical and chemical properties of fresh BB soot than processed soot, so our results were put in the context of fresh BB soot observations.

COMMENT: Page 8 line 255: what is the logic behind to adjust 580-660 nm range, not 500-580 nm range?

AUTHORS RESPONSE-The authors acknowledge that some more experimental detail is needed regarding this topic. Our main reasoning for this was that data in the 580-660 nm had poorer S/N than data in the 500-580 nm range. This is due to the smaller reflectivity of the mirrors in that range. The values for extinction, scattering, and absorption cross sections were high in the 580-660 nm range for 300 nm particles and low for 400 nm particles but maintained the same slope. For the same-day run for both wavelength ranges, we found nearly the same values for the 400 nm particles. In all cases the SSA did not change significantly due to adjusting the extinction and scattering values. Measurements were done several times at different days and the results are consistent.

COMMENT: Page 10 line 329: Author's mentions Cedar have higher SSA at flaming state than the smoldering stage for 300 nm particles. What is the [reason] behind this? Do authors want to say organics absorb more than black carbon in 500-660 nm? Since more BC is produced during flaming and more OC is produced during smoldering (Ward et al., 1992).

[Figure]

AUTHORS RESPONSE-Only 300 nm diameter particles exhibited a higher SSA in flaming stage than smoldering stage. Several others had SSA values that were indistinguishable between stages. The reviewer is correct, in that more discussion is warranted for these findings, and revisions to the paper will reflect that. Given that we do not have insight into the relative contributions of OC and EC, saying that OC is more absorbing than EC would be pure conjecture. While the authors agree with the conclusions of Ward et al. and many others, it is important to recognize that their observations were done over the entire size distribution, or at least the entirety of PM2.5. It has been shown that the smoldering phase emits larger, higher SSA particles (Reid et al., 2005). It is likely that at small particle diameters, such as 300 nm, EC has a greater contribution to than OC, giving rise to lower SSA values for the smoldering stage of this fuel at this size. To make up for this, larger particles could have a greater contribution of OC, resulting in its greater abundance in PM2.5. This is consistent with the larger observed SSA values for larger diameter particles in the smoldering stage of cedar combustion.

COMMENT: Page 10 line 351-354: Author's mentions there was no observable trend of SSA as a function of particle size and again wrote SSA diverged significantly at larger diameters. What do these sentences signify? And in Line 353 they mention particle size clearly plays the major role in determining the scattering or absorption properties. How that does not affect the SSA?

AUTHORS RESPONSE-It is not to say that there are not differences as a function of particle size, but there lacks a trend for the flaming stage. For the smoldering stage, as noted earlier in that paragraph, SSA increased with particle size for each fuel investigated. For the flaming stage, there is no systematic change of SSA with particle size.

COMMENT: Page 11 Line 394: Author's concluded that due to lack of variability in their SSA values for different wavelength the MCE of the burns for their work is < 0.92 based on Liu et al. (2014) study. If so, then why they called flaming for such burns? Based on Yokelson et al. (1996) definition, MCE of 0.9 represents roughly equal amount of

flaming and smoldering and MCE ∼0.8 are pure smoldering.

AUTHORS RESPONSE-Liu's results show that SSA varies strongly with fire-integrated modified combustion efficiency (MCE)—higher MCE results in lower SSA values and greater spectral dependence of SSA. SSA values between 0.6 and 0.8 correspond to MCF values between 0.9-0.95 for all samples at  = 405nm, 532 nm and 781nm. As the reviewer indicated for pine for example for flaming MCE = 0.990, for smoldering MCE =0.835, and MCE =0.96 for total fire integrated (Yokelson et al. 1996). Other earlier measurements are also consistent with these results, For example Reid et al. (2005) defined as MCE > 0.9 for flaming combustion, MCE < 0.9 for smoldering combustion and McMeeking, et al. (2009) had MCE = 0.80 smoldering phase of the fire and MCE = 0.99 for the flaming phase. Our conclusions were based on the most recent results of Liu et al. There is no statement on line 394 suggested by the reviewer calling it a flaming burn.

Reid, J. S., R. Koppmann, T. F. Eck, and D. P. Eleuterio (2005), A review of biomass burning emissions part II: Intensive physical properties of biomass burning particles, Atmos. Chem. Phys., 5, 799 – 825. Gavin R. McMeeking et al. (2009), Emissions of trace gases and aerosols during the open combustion of biomass in the laboratory; Journal of Geophysical Research 114, D19210 doi:10.1029/2009JD011836,

COMMENT: Page 12 Line 398-400: Author's mention that higher SSA values in field measurements than their observation suggest that the MCE values of wildfires are higher than controlled laboratory burns. They did not explain why burn with higher MCE could have higher SSA values. It is clear from laboratory studies that burns with higher MCE have lower SSA (Pokhrel et al., 2016; Liu et al., 2014; McMeeking et al., 2014) supporting the fact that more BC will produce during flaming stages of burns (Ward et al., 1992).

AUTHORS RESPONSE-The reviewer is correct. We will amend the text, stating "This could suggest that the MCE values of wildfires are lower than controlled laboratory

burns." This is likely due to the lower abundance of oxygen in wildfires, and the revised text will mention this as well. Yucatan Peninsula results will also be mentioned earlier in the paragraph. The remainder of the text stands.

COMMENT: Page 12 Line 408-410: Author's mentions, despite of having similar burn conditions with Hopkins et al. (2007), they found different SSA values for white pine than that of ponderosa pine needles/twigs. But they do not mention how they compare the burning conditions because they don't have either BC/OA or MCE calculations in their study.

AUTHORS RESPONSE-The authors are using the phrase "similar burning conditions" fairly loosely. In Hopkins et al. the fuel was simply burned on a platform. While this work was employed a burning drum, air flow was not especially restricted. This is opposed to other significantly different burning conditions, such as the reduced oxygen conditions of a forest fire or the high efficiency combustion done in a cook stove.

COMMENT: Throughout the document, authors wrote soot. Do they want to say only soot produced during biomass burning?

AUTHORS RESPONSE-The authors recognize that discussion is limited to soot produced from biomass burning (BB), and will make a statement at the end of our introduction regarding this. There are some discussions that apply to soot more generally from other sources (diesel exhaust, acetylene flames, etc.). We will review our use of the word "soot" throughout the document and specify BB soot if it aids in understanding and reduces confusion. However as pointed out by Buseck et al. (2012), there still exists ambiguity on the definitions of soot, black carbon, and carbonaceous aerosols.

Buseck, P. R.; Adachi, K.; Gelencsér, A.; Tompa, É.; Pósfai, M., Are black carbon and soot the same? Atmos. Chem. Phys. Discuss. 2012, 12 (9), 24821-24846.

Other things: We will remove the words "or mineral dust" from line 387 and add to the phrasing of the next sentence.

---

## Referee Comment (RC1) · Anonymous Referee #1 · 15 Jul 2016

The authors present very valuable data, regarding the optical properties of BB aerosols. This data is of high importance to both modelers and experimentalists. The authors did a careful job in providing an accurate measurement of the SSA, based on their previous work Singh et al., 2014 . However, the optical properties of these particles should be measured in a meaningful way. The authors should have a clear characteristic for the measured particles, in order to compare their results to the literature, or make any statement about the SSA. While, the SSA measurements are

carefully conducted, the conditions of the measurements are very poorly characterized
My recommendation, is to carefully address the comments listed below:

General comments:

1) The author should go over and recheck every single reference that they site, and make sure it is sited correctly. The introduction include at least 5 references which were sited either wrong or in a misleading way. P3 line 105-109: This sentence implies that, Riziq et al., 2007;Riziq et al., 2008 and Dinar et al., 2008 were using extinction minus scattering technique. Or alternatively measuring both extinction and scattering. These 3 papers where measuring only extinction using the CRD, and retrieved the scattering/abortion via Mie theory calculation. Same comment applies to the flowing 2 references Butler et al., 2007;Miller and Orr-Ewing, 2007. The authors claim that these papers measure the SSA of isolated aerosol particles. However, both the Butler et al., 2007 and the Miller and Orr-Ewing, 2007 were measuring/determining only the extinction of light by single aerosol particles.

Additionally, this paragraph is almost an exact copy of a paragraph in the group's previous paper (Singh at al., 2014 aerosol science and Technology, page 1345 last paragraph). Same comment for the paragraph stating at P4 line 132, is almost identical to a paragraph in p 1346 in Singh at al., 2014). The authors should address these issues.

P12 line 416-418: The Beyersdorf, 2013 reference is not a peer reviewed paper, this data was presented after the flight and represents very preliminary data from a meeting. The authors should remove this reference.

2) P6 lines 202-204: The authors state that, the samples were diluted and sonicated prior the introduction to CRDS and nephelometer, they also report that the size distribution changed after nebulization. Was this size distribution change a result of just atomizing problems, or is it a change in morphology or/and chemistry? The authors need to address this question as part of this paper's framework. The authors claim that their measurement would represent fresh soot that has undergone cloud processes,

however sonication and atomization may or may not, change the soot's morphology. If the authors wish make this statement, it should be supported by measurement (e.g. electron microscopy). Chemical analysis is also required to make any statement about these measurement. Does these measurement represent coated particle or un-coated aerosols? The authors clearly state in p15 lines 537-538, that composition and morphology, have the most significant effect on fresh BB aerosols. While, these properties are indeed dependent on the burning stage particle size fuel type and condition; one needs to show that these properties do not change in water solution and atomization/ solicitor. The authors mention that a future work will be addressing changes in the mixing state and morphology, these issues should be addressed as part of the current manuscript.

3) When comparing to their results to literature values, the authors refer to the MCE as an explanation for agreement/disagreement with the literature values. For example: 1) p11 line 394, 2) P13 line 436. However, the MCE was not measured in this work. This makes the compression to literate be very speculative. Also how could the authors show difference between fuel types without making sure that similar MCE is shown for all cases? I agree with P.Pokhrel comment claiming that:" They concluded burn condition does not control the SSA and AAE and mention as a one of the major findings in abstract but have not tested SSA and AAE correlation with either MCE or BC/OA ratio' The authors responded that they are currently working to address these important aspects of BB aerosols, including chemical analysis and optical properties as a function of aging using an indoor smog chamber in forthcoming work, this should be done (at least partially) as part of this manuscript framework

Same comment applies to the any of the possible explanations given in the last paragraph of p 13 (lensing, volatilize low molecular organics.) This explanations should be supported by some chemical analysis

Minor comments: 1) The author could address the multiple charged particles issue, by preforming a Multiple charge corrections (see for example Flores et al.,2012 ACP)

[Figure]

2) For all of the figures: please make sure all of the figure are consistent and clear. For example: The font size is different in every single figure. Figure 1: The drawing is cut on the left, figure 11: there is an axis on the right size (but other figure are open). Comparing figure 9 to 10: There have the same x –axis but one start with the actual number 500 and doesn't have minor ticks. Figure 11: the number 0.6 is cut, the legend is 'smooshed' etc. 3) Please provide an explanation to changes of the SSA with the particle size, for example: why is the Cedar smoldering 500nm has a lower SSA than the cedar smoldering 400nm?

Specific Comments: 1) Figure 4: This figure is very confusing and difficult to follow, Please make it more clear. 2) P8 line 255: Please change the period to comma. Avoid using 'and' after the period 3) P9 line 308: please add reference 4) P9 line 313: "particles that have the same electrical mobility, but different mobility diameters were separated" This sentence is not clear did the authors mean: same electrical mobility but different mass selection? 5) P9 lines 321-323: The authors state the SSA has a slope of zero over the range of 500-680nm, however the x-axis in the figure ends at 660nm. Please add the missing data to the figure. 6) P12 line 405: please change Lewis to Lewis et al., 7) P12 line 406: please add reference

---

## Author Comment (AC2) · 22 Jul 2016

REFEREE COMMENTS: The author should go over and recheck every single reference that they site, and make sure it is sited correctly. The introduction includes at least 5 references which were sited either wrong or in a misleading way. P3 line 105-109: This sentence implies that, Riziq et al., 2007;Riziq et al., 2008 and Dinar et al., 2008 were using extinction minus scattering technique alternatively measuring both extinction and scattering. These 3 papers where measuring only extinction using the

CRD, and retrieved the scattering/abortion via Mie theory calculation. Same comment applies to the flowing 2 references Butler et al., 2007; Miller and Orr-Ewing, 2007. The authors claim that these papers measure the SSA of isolated aerosol particles. However, both the Butler et al., 2007 and the Miller and Orr-Ewing, 2007 were measuring/determining only the extinction of light by single aerosol particles. Additionally, this paragraph is almost an exact copy of a paragraph in the group's previous paper (Singh at al., 2014 aerosol science and Technology, page 1345 last paragraph).

AUTHORS RESPONSE: We thank the referee for pointing the errors in the references: we are checking all the references to make sure citations are relevant and appropriate in the final manuscript.

Regarding Page 3 line 105-109, the authors don't feel this at all implies what the referee concluded. We only stated that they used the extinction minus scattering technique, but we didn't state they measured both extinction and scattering. However, to avoid confusion and in response to the comment the paragraph will be modified in the revised manuscript, and will read as:

"The extinction-minus-scattering technique has been used for airborne ambient measurements (Hallar et al. 2006), for studies involving optical properties of biomass aerosols and humic- like aerosols using CRD for measurement of extinction and Mie theory calculations to determine scattering and absorption (Riziq et al. 2007, 2008; Dinar et al. 2008), and for determining the extinction of isolated aerosol particles (Butler et al. 2007; Miller and Orr-Ewing 2007)."

REFEREE COMMENTS: Same comment for the paragraph stating at P4 line 132-140, is almost identical to a paragraph in p 1346 in Singh at al., 2014). The authors should address these issues.

AUTHORS RESPONSE We will revise the paragraph and refer the readers to the earlier paper for details. The paragraph will be revised in the manuscript and will read as "Details of the experimental method and derivation of key equations for particle optical

properties and CRDS analysis have been described by Singh et al. (2014) and references therein. Here, we only summarize the main points and encourage the reader to see the reference cited for details. The key equation for CRDS measurement is the extinction coefficient $\alpha$ext (m-1) is defined by Equation 1

Where cair is the speed of light in air and RL is the ratio of mirror-to-mirror distance d to the length of the cavity occupied by the sample, resulting in a unitless value >1. The ring-down time is $\tau$0 for an empty cavity and $\tau$ in the presence of a sample. Extinction coefficient is the product of the cross section ïA˛sext (m2/particle) and number density of particles NCRD (particles/cm3) in the CRD cavity. A unit conversion factor has been omitted for simplicity." REFEREE COMMENTS: P12 line 416-418: The Beyersdorf, 2013 reference is not a peer reviewed paper, this data was presented after the flight and represents very preliminary data from a meeting. The authors should remove this reference.

AUTHORS RESPONSE The authors are aware that this was a preliminary meeting presentation, but would like to keep the reference and change the text in the manuscript as below:

"In preliminary data presented in a meeting following NASA measurements during SEAC4RS, involving in situ sampling of the smoke from the Yosemite Rim Fire, the initial SSA of smoke was 0.92 and increased in the first 0–7 hrs. to 0.96, and was nearly constant after that (up to two days)." In the references it is cited as a meeting presentation. Sadly, no Rim Fire results on aerosol optical properties have been published from a peer-reviewed source; the closest being a publication that focused entirely on molecular emissions from the Rim Fire (doi 10.1016/j.atmosenv.2015.12.038).

REFEREE COMMENTS: P6 lines 202-204: The author's state that, the samples were diluted and sonicated prior the introduction to CRDS and nephelometer, they also report that the size distribution changed after nebulization. Was this size distribution change a result of just atomizing problems, or is it a change in morphology or/and

Interactive
comment

chemistry? The authors need to address this question as part of this paper's framework. The authors claim that their measurement would represent fresh soot that has undergone cloud processes, however sonication and atomization may or may not, change the soot's morphology. If the authors wish to make this statement, it should be supported by measurement (e.g. electron microscopy). Chemical analysis is also required to make any statement about this measurement. Does this measurement represent coated particle or un-coated aerosols? The authors clearly state in p15 lines 537-538, that composition and morphology, have the most significant effect on fresh BB aerosols. While, these properties are indeed dependent on the burning stage particle size fuel type and condition; one needs to show that these properties do not change in water solution and atomization/ solicitor. The authors mention that a future work will be addressing changes in the mixing state and morphology, these issues should be addressed as part of the current manuscript.

AUTHORS RESPONSE We believe the change in the size distribution is mainly a result of an atomizing problem though change in morphology cannot be ruled out. The re-aerosolized particles will likely have a near spherical core-shell morphology. The samples were kept in distilled water and not exposed to the environment. As we stated in our previous response to a similar comment, we have several reasons for comparing our values to fresh BB soot. While some changes in the particle size distribution were observed upon impingement, this sampling scheme (or, really, any offline sampling scheme) is unlike any natural atmospheric processing. No photochemical changes were allowed to take place. The system would be too dilute for SOA formation via acid chemistry. Semi-volatile species would be almost immediately put into the condensed phase, though solubility would be a more important factor in determining their interactions with the particulate phase. As indicated in our previous response, further clarification on these issues will be included in the revision.

Addressing changes in morphology upon impingement and re-aerosolization, along with changes due to atmospheric aging, is no small task. We are in the process of

building an indoor smog chamber. The characterization of this chamber and conducting optical and chemical properties measurements will probably take another year to address, and will result in at least two distinct publications. For considerations of time, publication length, and narrative flow, the authors do not believe these requests are feasible for the current manuscript. The authors are more than willing to revisit the conclusions of this paper when future results become available.

REFEREE COMMENTS: When comparing to their results to literature values, the authors refer to the MCE as an explanation for agreement/disagreement with the literature values. For example: 1) p11 line 394, 2) P13 line 436. However, the MCE was not measured in this work. This makes the [comparison] to [literature values] be very speculative.

AUTHORS RESPONSE The authors are examining their results in light of what is currently known about SSA and AAE of BB aerosols. Not to do this would be negligent. Specifically, there are schemes that relate SSA and AAE to either MCE or EC/(EC+OC). If these schemes are robust, new data should also fit within their trends. The authors disagree that this comparison is speculative. Indeed, this is, essentially, a "two equations, one unknown" problem. The two equations are trends (relationships) of SSA with MCE and AAE with MCE (or with EC/(EC+OC) in Pokhrel's case). The unknown, MCE or EC/(EC+OC), can be solved precisely.

REFEREE COMMENTS: Also how could the authors show difference between fuel types without making sure that similar MCE is shown for all cases?

AUTHORS RESPONSE While the authors are attempting that level of control in future work, no work to date has tried to control MCE. But what gives rise to differences in MCE? The authors state, in the manuscript, that it is influenced by fuel type, fuel state, and burning conditions. Examples can be found on line 27 of the abstract and line 524. It is likely that MCE is varying with fuel type, and that these are not independent variables.

[Figure]

REFEREE COMMENTS: I agree with P.Pokhrel comment claiming that: "They concluded burn condition does not control the SSA and AAE and mention as a one of the major findings in abstract but have not tested SSA and AAE correlation with either MCE or BC/OA ratio." The authors responded that they are currently working to address these important aspects of BB aerosols, including chemical analysis and optical properties as a function of aging using an indoor smog chamber in forthcoming work, this should be done (at least partially) as part of this manuscript framework.

AUTHORS RESPONSE: The authors, in their response, went on to state that at no time did we conclude that combustion conditions and MCE have no effect on SSA or AAE. If the reviewer could point out where in the manuscript this was stated (either explicitly or implied), the authors will rectify this misunderstanding. However, the authors are unable to find such an occurrence.

The sample at issue is mainly white pine. It exhibits the following properties, based on Figures 1 and 4 of Pokhrel et al.:

1. The AAE is high, suggesting EC/(EC+OC) and MCE are low 2. SSA values are low, suggesting EC/(EC+OC) and MCE are high 3. There is a lack of SSA spectral dependence, suggesting EC/(EC+OC) and MCE are low Even if the authors were to measure MCE or EC/(EC+OC) for white pine, it cannot have an MCE both above and below 0.92 or an EC/(EC+OC) both above and below 0.2. It is either at the high end or the low end where the relationships of Liu et al. and Pokhrel et al. do not work well with our white pine observations.

That being said, further explanation is clearly warranted in the manuscript. The authors would have liked to provide a graphical comparison of our results with FLAME-4 results, but we were unable because Liu et al. did not provide their raw data used in their figures. However, Pokhrel et al., thanks to the encouragement of ACPD, did provide their data. While Liu et al. and Pokhrel et al. plot SSA and AAE against MCE, EC/OC, and EC/(EC+OC), neither publication plots SSA against AAE. We have done so here

and Figure 1 and accompanying discussion will be included in the revised manuscript:

"As can be seen in Figure X, many of our measurements inhabit a distinct location in AAE/SSA space. The AAE is higher and the SSA is lower than most FLAME-4 observations. Part of this difference may be that previous measurements were done for the entire burn and all diameters below 2.5 $\mu$m, whereas measurements in this work were segregated by size and burning stage. When both data sets were combined and fit to a power law function, the y-offset increased and the fit had greater power dependence."

REFEREE COMMENTS: Same comment applies to the any of the possible explanations given in the last paragraph of p 13 (lensing, volatilize low molecular organics.) This explanation should be supported by some chemical analysis

AUTHORS RESPONSE: As stated in the last sentence, the aim of that paragraph was not truly an explanation of results. Indeed, we don't discuss our results at all in the paragraph. We are exercising due diligence, and mentioning potentially important effects that we could not constrain at the time this data was gathered. We will attempt to clarify this point, earlier in the paragraph, in the revised manuscript.

REFEREE MINOR COMMENTS: 1) The author could address the multiple charged particles issue, by performing a multiple charge corrections (see for example Flores et al., 2012 ACP)

AUTHORS RESPONSE The authors would like to thank the reviewer for reminding us of this factor. In earlier work on monodisperse polystyrene, it was not an important factor. While it's certainly possible for the authors to calculate the contribution of multiply charged particles, it is not altogether clear how it would be used. Flores et al. and many others need to account for particle size because of their use Mie theory fitting to determine RI. For 300 nm particles, the SSA of 400 nm particles could potentially approximate to the 424 nm particles that would have double the geometric cross section with twice the charge. However, there is not enough available information (measurements of SSA and AAE at 566 and 707 nm diameters) to adjust the 400 or 500 nm particles. We will mention this in the "due diligence" paragraph, and will discuss its potential impact for the 300 nm particle case, where such an assessment can be made. Most troubling is the mention of "errata" by Flores et al. concerning the original paper by Wiedensohler, even though the authors were unable to locate any such errata.

REFEREE COMMENTS: 2) For all of the figures: please make sure all of the figures are consistent and clear. For example: The font size is different in every single figure. Figure 1: The drawing is cut on the left, figure 11: there is an axis on the right size (but other figure are open).Comparing figure 9 to 10: There have the same x –axis but one start with the actual Number 500 and doesn't have minor ticks. Figure 11: the number 0.6 is cut, the legend is 'smooshed' etc.

AUTHORS RESPONSE The figures will be corrected to address the concerns in the revised manuscript.

REFEREE COMMENTS: 3) Please provide an explanation to changes of the SSA with the particle size, for example: why is the Cedar smoldering 500nm has a lower SSA than the cedar smoldering 400nm?

AUTHORS RESPONSE We did respond to the similar comment by P. Pokhrel. It is likely that at small particle diameters, such as 300 nm, EC has a greater contribution to the particle mass than OC, giving rise to lower SSA values for the smoldering stage of this fuel at this size. To make up for this, larger particles could have a greater contribution of OC, resulting in its greater abundance in PM2.5. This is consistent with the larger observed SSA values for larger diameter particles in the smoldering stage of cedar combustion.

SPECIFIC COMMENTS AND RESPONSES: 1) Figure 4: This figure is very confusing and difficult to follow, please make it more clear.

AUTHORS RESPONSE The figure will be modified for clarity and the caption will be

Interactive
comment

more descriptive in the revised manuscript as below:

"Figure 4- The flow of the calculation for determining average values of $\sigma$ext, $\sigma$scat, $\omega$, and their errors. Variables with an asterisk represent individual measurements. $\sigma$ext for each experiment is derived from the $\alpha$ext and the number density within the cavity, via Equation 1. This number density is found using Equation set 3. The standard deviation of $\sigma$ext for each experiment is found using Equation 2. The RSD of $\sigma$ext for each experiment is found, averaged, and multiplied by the average $\sigma$ext to get the average standard deviation of $\sigma$ext. $\alpha$scat for each experiment is corrected and $\sigma$scat is found using the number density in the nephelometer. This is averaged and its standard deviation found from the run-to-run variability of $\sigma$scat, the RSD of NNeph, and the correction factor error. The RSD of $\alpha$scat is based on the run-to-run variability of $\alpha$scat and the correction factor error. The SSA of each run is based on $\alpha$scat and $\alpha$ext for each run and the number density relationship. This is averaged, and the run-to-run variability of SSA determined. This variability is used, along with the RSD of $\alpha$scat and $\alpha$ext, to determine the SSA error."

2) P8 line 255: Please change the period to comma. Avoid using 'and' after the period.

Change will be made in the manuscript.

3) P9 line 308: please add reference

The reference is added.

4) P9 line 313: "particles that have the same electrical mobility, but different mobility diameters were separated" This sentence is not clear did the authors mean: same electrical mobility but different mass selection?

This statement is correct. Mass of the particles is proportional to the product of the electrical mobility and mobility diameter and inversely proportional to the drag coefficient, which is also a function of the mobility diameter.

5) P9 lines 321-323: The authors state the SSA has a slope of zero over the range of

500-680nm, however the x-axis in the figure ends at 660nm. Please add the missing data to the figure.

The data was taken from 500-660 nm, 680 was an error. It will be corrected.

6) P12 line 405: please change Lewis to Lewis et al.,

This is done.

7) P12 line 406: please add reference

This was the same reference cited on line 405.

Figure Caption

Figure 1: Results of this work compared to FLAME-4 results (Liu et al, 2013; Pokhrel et al., 2016). A power law fit was performed in the form of AAE = a + b•SSAc was performed for FLAME-4 and combined data. For FLAME-5, a = 2.402±0.296, b = 5.298±0.587, and c = 28.53±8.42. For the combined data set, a = 2.852±0.187, b = 4.961±0.599, and c = 36.965±11.300.

[Figure]

[Figure]

**Fig. 1.** Figure 1

$$\alpha_{ext} = \frac{R_L}{c_{air}}\left(\frac{1}{\tau} - \frac{1}{\tau_0}\right) = \sigma_{ext}N_{CRD}$$

**Fig. 2.** Equation 1

---

## Referee Comment (RC2) · Anonymous Referee #2 · 10 Aug 2016

The manuscript deals with an interesting and tricky topic, providing very useful data and information on optical properties of BB-derived particles. I would like to congratulate the Authors for the huge quantity of work done and for the attention dedicated to solve all the technical aspects related to this kind of measurements. From the linguistic point of view, the paper is quite clear and well written. I see in this manuscript two main weaknesses. The first one regards two important lacks: the calculation of the Modified Combustion Efficiency (MCE) and the determination of EC and OC. In this field

of study, this information is very useful since both influence the final optical properties of the particles. Although these lacks don't affect the goodness of the results, they make impossible a direct comparison between the data they show and the literature they cite, forcing the authors to a sort of speculation (as pointed out by the other Referees). In the Authors response to AC1 they state that "there are schemes that relate SSA and AAE to either MCE...the unknown, MCE or EC/(EC+OC), can be solved precisely" knowing SSA and AAE. But just few lines later they state "But what gives rise to differences in MCE? The authors state, in the manuscript, that it is influenced by fuel type, fuel state, and burning conditions". So, if the authors would calculate the MCE or EC/(EC+OC) values considering the schemes available in literature, they are assuming that fuel type, fuel state and burning conditions are the same in both the experiments. How it could be possible? The second one is related on the "distance" between the BB aerosols produced in the Authors "soot generation setup" and the particles they are measuring. They clearly state that particles changed in size distribution and morphology after the various processes of collection, sonication, nebulization. Also chemical composition changed both during preparation (partial removal of semi-volatile species) and during storage (moreover Authors do not determine chemical composition in any way). Although I agree with the authors that the particles they are measuring are likely more close to fresh than to aged BB aerosols (no photochemical transformation, no SOA formation), these particles are very different from the original ones. I wonder how much the optical properties shown in this paper are representative of real fresh BB particles. I think that the previous Referees have pointed out the crucial problems and I have no questions to add, except one: in Figg. 5-10 there is a clear point of discontinuity (especially in Figg. 6, 8 and 10) in correspondence of 580 nm: the values measured with the dye laser (< 580nm) are more similar for the different fuels while much more widespread in the case of the OPO laser (>580 nm). I have not find any comment in the text about this evident difference. The authors are aware of the limitations present in their work. I think that these limitations are well explained in the text and clear to the reader. The Authors should anyway include some integrations as suggested by the

Referees. Overall, I consider this paper scientifically remarkable and complete.

---

## Author Comment (AC3) · 15 Aug 2016

REFEREE COMMENTS: The first one regards two important lacks: the calculation of the Modified Combustion Efficiency (MCE) and the determination of EC and OC. In this field of study, this information is very useful since both influence the final optical properties of the particles. Although these lacks don't affect the goodness of the results, they make impossible a direct comparison between the data they show and the literature they cite, forcing the authors to a sort of speculation (as pointed out by the

other Referees).

AUTHORS RESPONSE:

The authors admit that this additional information on MCE would be useful, and is being implemented in our future work, which is currently in progress. In this work the burning conditions will be highly controlled for temperature and oxygen content, allowing us to vary MCE. While no direct MCE- or EC/(EC+OC)-based comparisons are possible, the discussion uses qualitative comparisons of burn conditions. At this time, the only possibility of performing these measurements would be to return to the original fuel samples and measure their fire-integrated CO and $CO_2$ values. However, even if those fuels were still there, they would have been sitting in the open for over a year. The fidelity of the samples would be very questionable.

REFEREE COMMENTS: In the Authors response to AC1 they state that "there are schemes that relate SSA and AAE to either MCE. . .the unknown, MCE or EC/(EC+OC), can be solved precisely" knowing SSA and AAE. But just few lines later they state "But what gives rise to differences in MCE? The authors state, in the manuscript, that it is influenced by fuel type, fuel state, and burning conditions". So, if the authors would calculate the MCE or EC/(EC+OC) values considering the schemes available in literature, they are assuming that fuel type, fuel state and burning conditions are the same in both the experiments. How it could be possible?

AUTHORS RESPONSE:

The MCE- and EC/(EC+OC)-based schemes used data gathered during FLAME-4 experiments. In that work, a variety of fuels were burned under several conditions (mainly open burns and several types of cookstoves). Both of these factors are already varying, and the papers of Pokhrel et al. and Liu et al. attempt to find a robust fit for SSA and AAE as a function of MCE and EC/(EC+OC), respectively. This is a worthy goal, especially for use in modeling efforts. However, for some samples we have investigated, these trends have some deficiencies.

REFEREE COMMENTS: The second one is related on the "distance" between the BB aerosols produced in the Authors "soot generation setup" and the particles they are measuring. They clearly state that particles changed in size distribution and morphology after the various processes of collection, sonication, nebulization. Also chemical composition changed both during preparation (partial removal of semi-volatile species) and during storage (moreover Authors do not determine chemical composition in any way). Although I agree with the authors that the particles they are measuring are likely more close to fresh than to aged BB aerosols (no photochemical transformation, no SOA formation), these particles are very different from the original ones. I wonder how much the optical properties shown in this paper are representative of real fresh BB particles.

AUTHORS RESPONSE:

While it is possible that these samples have more in common with soot that has undergone processing in pyrogenic clouds, the authors are not aware of any such field measurements. Thus, putting our measurements in that context is not currently possible. The closest comparison would be fresh soot.

REFEREE COMMENTS: I think that the previous Referees have pointed out the crucial problems and I have no questions to add, except one: in Figg. 5-10 there is a clear point of discontinuity (especially in Figg. 6, 8 and 10) in correspondence of 580 nm: the values measured with the dye laser (< 580nm) are more similar for the different fuels while much more widespread in the case of the OPO laser (>580 nm). I have not found any comment in the text about this evident difference.

AUTHORS RESPONSE:

Two sets of mirrors used in this work and 580 nm marks the boundary between the ranges at which they are highly reflective. It does not denote the wavelength range of the two light sources. Due to differences in mirror reflectivity, differences in the error and level of noise are apparent in the different ranges. All the work was done

using OPO. The dye laser was not used for this work. The experimental section on the paper will include a sentence to show that only the OPO was used. Regarding the discontinuity at 580 nm, we already provided an explanation in the text Line 246-259. In response to comments by Rudra Pokhrel on the same issue, we provided the following explanation: "Our main reasoning for this was that data in the 580-660 nm had poorer S/N than data in the 500-580 nm range. This is due to the smaller reflectivity of the mirrors in that range. The values for extinction, scattering, and absorption cross sections were high in the 580-660 nm range for 300 nm particles and low for 400 nm particles but maintained the same slope. For the same-day run for both wavelength ranges, we found nearly the same values for the 400 nm particles. In all cases the SSA did not change significantly due to adjusting the extinction and scattering values. Measurements were done several times at different days and the results are consistent" It is also worth noting that the level of noise is not the same for different particle sizes, which is largely due to number density differences.

REFEREE COMMENTS: The authors are aware of the limitations present in their work. I think that these limitations are well explained in the text and clear to the reader. The Authors should anyway include some integrations as suggested by the Referees. Overall, I consider this paper scientifically remarkable and complete.

AUTHORS RESPONSE:

The authors would like to thank the referee for their kind remarks. We are unsure about what 'integrations' the reviewer is referring to. If the referee is suggesting that we integrate the comments offered by the other referees, we have already indicated how we intend to integrate their suggestions into the final text for publication.